

**Effects of elevated CO$_2$ and temperature on phytoplankton community**
**biomass, species composition and photosynthesis during an autumn**
**bloom in the Western English Channel**
Matthew Keys[1, 2], Gavin Tilstone[1*], Helen S. Findlay[1], Claire E. Widdicombe[1] and Tracy Lawson[2].
[1] Plymouth Marine Laboratory, Prospect Place, The Hoe, Plymouth, PL1 3DH, UK.
[2] University of Essex, Wivenhoe Park, Colchester, CO4 3SQ, UK.
*Correspondence to*: G. Tilstone (ghti@pml.ac.uk)
**Abstract**
The combined effects of elevated pCO$_2$ and temperature were investigated during an autumn
phytoplankton bloom in the Western English Channel (WEC). A full factorial 36-day microcosm
experiment was conducted under year 2100 predicted temperature (+ 4.5 °C) and pCO$_2$ levels
(800 µatm). The starting phytoplankton community biomass was 110.2 (± 5.7 sd) mg carbon (C)
m$^{-3}$ and was dominated by dinoflagellates (~50 %) with smaller contributions from
nanophytoplankton (~13 %), cryptophytes (~11 %)and diatoms (~9 %). Over the experimental
period total biomass was significantly increased by elevated pCO$_2$ (20-fold increase) and
elevated temperature (15-fold increase). In contrast, the combined influence of these two
factors had little effect on biomass relative to the ambient control. The phytoplankton
community structure shifted from dinoflagellates to nanophytoplankton at the end of the
experiment in all treatments. Under elevated pCO$_2$ nanophytoplankton contributed 90% of
community biomass and was dominated by *Phaeocystis* spp., while under elevated temperature
nanophytoplankton contributed 85% of the community biomass and was dominated by smaller
nano-flagellates. Under ambient conditions larger nano-flagellates dominated while the smallest
nanophytoplankton contribution was observed under combined elevated pCO$_2$ and temperature
(~40 %). Dinoflagellate biomass declined significantly under the individual influences of
elevated pCO$_2$, temperature and ambient conditions. Under the combined effects of elevated
pCO$_2$ and temperature, dinoflagellate biomass almost doubled from the starting biomass and
there was a 30-fold increase in the harmful algal bloom (HAB) species, *Prorocentrum cordatum*.
Chlorophyll a normalised maximum photosynthetic rates (P$^B_m$) increased > 6-fold under
elevated pCO$_2$ and > 3-fold under elevated temperature while no effect on P$^B_m$ was observed
when pCO$_2$ and temperature were elevated simultaneously. The results suggest that future
increases in temperature and pCO$_2$ do not appear to influence coastal phytoplankton



productivity during autumn in the WEC which would have a negative feedback on atmospheric
$CO_2$.

### 1. Introduction

Oceanic uptake of atmospheric $CO_2$ has increased by ~42% over pre-industrial levels, with an
on-going annual increase of ~0.4%. Current $CO_2$ level has reached ~400 µatm and has been
predicted to rise to >700 µatm by the end of this century (Alley *et al.*, 2007), with estimates
exceeding 1000 µatm (Raupach *et al.*, 2007; Raven *et al.*, 2005). The oceans are absorbing $CO_2$
from the atmosphere, which results in a shift in oceanic carbonate chemistry resulting in a
decrease in seawater pH or 'Ocean Acidification' (OA). The projected increase in atmospheric
$CO_2$ and corresponding increase in ocean uptake, is predicted to result in  a decrease in global
mean seawater pH of 0.3 units below the present value of 8.1 to 7.8 (Wolf-gladrow et al., 1999).
Under this scenario, the shift in dissolved inorganic carbon (DIC) equilibria has wide ranging
implications for phytoplankton photosynthetic carbon fixation rates and growth (Riebesell,

46    2004).

Concurrent with OA, elevated atmospheric $CO_2$ and other climate active gases have warmed the
planet by ~0.6 °C over the past 100 years (IPCC, 2007). Atmospheric temperature has been
predicted to rise by a further 1.8 to 4 °C by the end of this century (Alley et al., 2007).
Phytoplankton metabolic activity may be accelerated by increased temperature (Eppley, 1972),
which can vary depending on the phytoplankton species and their physiological requirements
(Beardall and Stojkovic, 2006). Long-term data sets already suggest that ongoing changes in
coastal phytoplankton communities are likely due to climate shifts and other anthropogenic
influences (Edwards et al., 2006; Smetacek and Cloern, 2008; Widdicombe et al., 2010).  The
response to OA and temperature can potentially alter the community composition, community
biomass and photo-physiology. Understanding how these two factors may interact
(synergistically or antagonistically) is critical to our understanding and for predicting future
primary productivity (Boyd and Doney, 2002).
Laboratory studies of phytoplankton species in culture and studies on natural populations in
the field have shown that most species exhibit sensitivity, in terms of growth and
photosynthetic rates, to elevated $pCO_2$ and temperature individually. To date, only a few studies
have investigated the interactive effects of these two stressors on natural populations (e.g.
Coello-Camba et al., 2014; Feng et al., 2009; Gao et al., 2017; Hare et al., 2007). Most laboratory
studies have varied results with species-specific responses, for example, with the diatom
*Thalassiosira weissflogii,* $pCO_2$ elevated to 1000 µatm and + 5 °C temperature increase
synergistically enhanced growth, while the same conditions resulted in a reduction in growth



for the diatom *Dactyliosolen fragilissimus* (Taucher et al., 2015). Although there have been fewer
studies on dinoflagellates, similar variability in responses has been observed, e.g. (Errera et al.,
2014; Fu et al., 2008). In natural populations, elevated $pCO_2$ has stimulated growth in pico- and
nanophytoplankton communities (Engel et al., 2008) while increased temperature has reduced
biomass of these groups (Moustaka-Gouni et al., 2016). In a recent field study on natural
phytoplankton communities, elevated temperature (+ 3°C above ambient) enhanced community
biomass in natural populations but the combined influence of elevated temperature and $pCO_2$
caused a reduction in biomass (Gao et al., 2017).
Phytoplankton species composition, abundance and biomass has been measured at the time-
series station L4 in the western English Channel (WEC) since 1992, to evaluate how global
changes could drive future shifts in phytoplankton community structure and carbon
biogeochemistry. To compliment the biological time series, key environmental parameters for
monitoring the health and state of the WEC are measured weekly including depth profiles of
seawater temperature. Dissolved inorganic carbon (DIC) and total alkalinity (TA) has been
sampled at station L4 since 2008. Over the past 50 years a 0.5 °C warming has been observed in
the WEC (Smyth et al., 2010). The DIC and TA time series is relatively short and as such there is
no significant trend in the calculated $pCO_2$, although it has shown an increase.
Based on the existing literature, the working hypotheses of this study are that: (1) community
biomass will increase differentially under individual treatments of elevated temperature and
$pCO_2$; (2) elevated $pCO_2$ will lead to taxonomic shifts due to differences in species-specific $CO_2$
concentrating mechanisms and/or RuBisCO specificity; (3) photosynthetic carbon fixation rates
will increase differentially under individual treatments of elevated temperature and $pCO_2$; (4)
elevated temperature will lead to taxonomic shifts due to species-specific thermal optima; (5)
temperature and $pCO_2$ elevated simultaneously will have synergistic effects.
The objectives of the study were to investigate: 1) the combined effects of elevated $pCO_2$ and
temperature on phytoplankton community structure, biomass and photosynthetic carbon
fixation rates during the autumn transition from diatoms and dinoflagellates to
nanophytoplankton at station L4; 2) assess the natural variability in phytoplankton community
structure and the carbon biomass of the dominant species observed in the experimental
community relative to long-term observations at station L4 over two decades (1993-2014); and
3) assess the distribution of biomass of the dominant species observed at the end of the
experiment relative to the in-situ gradients of temperature and $pCO_2$ observed at station L4. The
effects of elevated $pCO_2$ and temperature on phytoplankton succession in autumn is presently
unknown.



## 2. Materials and methods

### 2.1 Time series, phytoplankton community composition

Station L4 (50° 15'N, 4° 13'W) is located 13 km SSW of Plymouth in a water depth of ~54 m (Harris, 2010) and is regarded as one of Europe's principal coastal time series sites. Sampling is conducted on a weekly basis (weather permitting) and has been on-going since 1992 (http://www.westernchannelobservatory.org). Phytoplankton taxonomic composition was enumerated from seawater samples collected from 10 m depth, fixed with 2 % (final concentration) acid Lugol's iodine solution and analysed by inverted light microscopy using the Utermöhl counting technique (Utermöhl, 1958; Widdicombe *et al.*, 2010). For phytoplankton carbon biomass values; taxa-specific mean cell bio-volumes were calculated following Kovala & Larrance, (1966) and converted to carbon using the equations of Menden-Deuer & Lessard, (2000).

### 2.2 Perturbation experiment, sampling and experimental set-up

Experimental seawater containing a natural phytoplankton community was sampled at station L4 on 7th October 2015 from 10 m depth (40 L). The experimental seawater was gently pre-filtered through a 200 μm Nitex mesh to remove zooplankton grazers, into two 20 L acid-cleaned carboys. In addition, 320 L of seawater was collected into sixteen 20 L acid-cleaned carboys from the same depth for use as experimental media. Immediately upon return to the laboratory the media seawater was filtered through an in-line 0.2 and 0.1 μm filter (Acropak™, Pall Life Sciences) then stored in the dark at 14 °C until use. The experimental seawater was gently and thoroughly mixed and transferred in equal parts from each carboy (to ensure homogeneity) to sixteen 2.5 L borosilicate incubation bottles (4 sets of 4 replicates). The remaining experimental seawater was sampled for initial (T0) concentrations of nutrients, chlorophyll *a*, total alkalinity, dissolved inorganic carbon, particulate organic carbon (POC) and nitrogen (PON) and was also used to characterise the starting experimental phytoplankton community. The incubation bottles were placed in an outdoor simulated in-situ incubation culture system and each set of replicates were linked to one of four 22 L reservoirs filled with the filtered seawater media. Neutral density spectrally corrected blue filters (Lee Filter no. 301) were placed between polycarbonate sheets and mounted to the top, sides and ends of the incubation system to provide ~50 % irradiance, approximating PAR measured at 10 m depth at station L4 on the day of sampling prior to starting experimental incubations. The media was aerated with $CO_2$ free air and 5 % $CO_2$ in air precisely mixed using a mass flow controller (Bronkhorst UK Limited) and used for the microcosm dilutions as per the following experimental design: (1) control (390 μatm $pCO_2$, 14.5 °C matching station L4 in-situ values),



(2) high temperature (390 µatm $pCO_2$, 18.5 °C), (3) high $pCO_2$ (800 µatm $pCO_2$, 14.5 °C) and (4)
combination (800 µatm $pCO_2$, 18.5 °C).
Initial nutrient concentrations (measured at 0.24 µM nitrate + nitrite, 0.086 µM phosphate and
2.14 µM silicate on 7th October 2015) were amended to 8 µM nitrate+nitrite and 0.5 µM
phosphate to provide favourable growth conditions. As the phytoplankton community was
sampled over the transitional phase from diatoms and dinoflagellates to nanophytoplankton,
the in-situ silicate concentration was maintained to reproduce the silicate concentrations
typical of this time of year (Smyth et al., 2010). Media transfer and sample acquisition was
facilitated by peristaltic pumps and semi-continuous daily dilution rates were set between 10-
13 % of the incubation bottle volume following 48 hrs acclimation in batch culture. $CO_2$
enriched seawater was added to the high $CO_2$ treatment replicates every 24 hrs, acclimating the
natural phytoplankton population to increments of elevated $pCO_2$ from ambient to ~800 µatm
over 8 days followed by maintenance at ~800 µatm as per the method described by Schulz *et al*,
(2009). This protocol was preferred since some phytoplankton species are inhibited by the
mechanical effects of direct bubbling (Riebesell et al., 2010; Shi et al., 2009) which can cause a
reduction in growth rates and the formation of aggregates (Love et al., 2016). pH was monitored
daily to adjust the $pCO_2$ of the experimental media (+/-) prior to dilutions to maintain target
$pCO_2$ levels in the incubation bottles.
**2.3 Analytical methods, experimental seawater**
**2.3.1 Chlorophyll *a***
Chlorophyll *a* (Chl a) was measured in each incubation bottle. 100 mL triplicate samples from
each replicate were filtered onto 25 mm GF/F filters (nominal pore size 0.7 µm), extracted in 90
% acetone overnight at -20 °C and chl *a* was estimated on a Turner Trilogy ™ fluorometer using
the non-acidified method of Welschmeyer (1994). The fluorometer was calibrated against a
stock Chl *a* standard (*Anacystis nidulans*, Sigma Aldrich, UK), the concentration of which was
determined with a Perkin Elmer™ spectrophotometer at wavelengths 663.89 and 750.11 nm.
Samples for Chl *a* were taken every 2-3 days.
**2.3.2 Carbonate system**
70 mL samples for total alkalinity (TA) and dissolved inorganic carbon (DIC) analysis were
collected from each experimental replicate, stored in amber borosilicate bottles with no head
space and fixed with 40 µL of super-saturated $Hg_2Cl_2$ solution for later determination (Apollo
SciTech™ Alkalinity Titrator AS-ALK2; Apollo SciTech™ AS-C3 DIC analyser, with analytical
precision of 3 µmol kg$^{-1}$). Duplicate measurements were made for TA and triplicate





measurements for DIC. Carbonate system parameter values for media and treatment samples
were calculated from TA and DIC measurements using the programme $CO_2$sys (Pierrot et al.,
2006) with dissociation constants of carbonic acid of Mehrbach *et al.*, (1973) refitted by Dickson
and Millero (Dickson and Millero, 1987). Samples for TA and DIC were taken every 2-3 days.
**2.3.3 Phytoplankton community analysis**
Phytoplankton community analysis was performed by flow cytometry (Becton Dickinson Accuri
™ C6) for the 0.2 to 18 μm size fraction following Tarran *et al.*, (2006) and inverted light
microscopy was used to enumerate cells > 18 μm (BS EN 15204,2006). For flow cytometry, 2
mL samples fixed with glutaraldehyde to a final concentration of 2 % were flash frozen in liquid
nitrogen and stored at -80 °C for later analysis. For inverted light microscopy, 140 mL samples
were fixed with 2 % (final concentration) acid Lugol's iodine solution and analysed by inverted
light microscopy (Olympus™ IMT-2) using the Utermöhl counting technique (Utermöhl, 1958;
Widdicombe *et al.*, 2010). Phytoplankton community samples were taken at T0, T10, T17, T24
and T36.
**2.3.4 Phytoplankton community biomass**
The smaller size fraction identified and enumerated through flow cytometry;
picophytoplankton, nanophytoplankton, *Synechoccocus*, coccolithophores and cryptophytes
were converted to carbon biomass (mg C m$^{-3}$) using a spherical model to calculate mean cell
volume:
$(\frac{4}{3} * \pi * r^3)$
and a conversion factor of 0.22 pg C μm$^{-3}$ (Booth, 1988). A conversion factor of 0.285 pg C μm$^{-3}$
was used for coccolithophores (Tarran et al., 2006) and cell a volume of 113 μm$^3$ and carbon
cell$^{-1}$ value of 18 pg applied for *Phaeocystis* spp. (Widdicombe *et al.*, 2010). *Phaeocystis* spp.
were identified and enumerated by flow cytometry separately to the nanophytoplankton class
due to high observed abundance in in the high $pCO_2$ treatment. Mean cell measurements of
individual species/taxa were used to calculate cell bio-volume for the 18 μm + size fraction
according to Kovala and Larrance (1966) and converted to biomass according to the equations
of Menden-Deuer & Lessard, (2000).
**2.3.5 POC and PON**
Samples for particulate organic carbon (POC) and particulate organic nitrogen (PON) were
taken at T0, T15 and T36.150 mL samples were taken from each replicate and filtered under
gentle vacuum onto pre-ashed 25mm glass fibre filters (GF/F, nominal pore size 0.7 μm). Filters





were stored in acid washed petri-slides at -20 °C until further processing. Sample analysis was
conducted using a Thermoquest Elemental Analyser (Flash 1112). Acetanilide standards (Sigma
Aldrich, UK) were used to calibrate measurements of carbon and nitrogen and also used during
the analysis to account for any drift in measured values.

### 2.3.6 Chl fluorescence-based photophysiology

Photosystem II (PSII) variable Chl fluorescence parameters were measured using a fast
repetition rate fluorometer (FRRf) (FastOcean sensor in combination with an Act2Run
laboratory system, Chelsea Technologies, West Molesey, UK). The excitation wavelengths of the
FRRf's light emitting diodes (LEDs) were 450, 530 and 624 nm. The instrument was used in
single turnover mode with a saturation phase comprising 100 flashlets on a 2 μs pitch and a
relaxation phase comprising 40 flashlets on a 50 μs pitch. Measurements were conducted in a
temperature controlled chamber at 15 °C. The minimum ($F_o$) and maximum ($F_m$) Chl
fluorescences were estimated according to Kolber et al., (1998). Maximum quantum yields of
PSII were calculated as:
$F_v / F_m = (F_m − F_o) / F_m$
PSII electron flux was calculated on a volume basis ($JV_{PSII}$; mol e$^-$ m$^{-3}$ d$^{-1}$) using the absorption
algorithm (Oxborough et al., 2012) following spectral correction by normalising the FRRf LED
emission to the white spectra using Fast$^{PRO}$ 8 software. This step required inputting the
experimental phytoplankton community fluorescence excitation spectra values (FES). Since we
did not measure the FES of our experimental samples, we used mean literature values for each
phytoplankton group calculated proportionally (based on percentage contribution to total
estimated biomass per phytoplankton group) as representative values for our experimental
samples. The $JV_{PSII}$ rates were converted to Chl specific carbon fixation rates (mg C (mg Chl $a$)$^{-1}$
m$^{-3}$ h$^{-1}$), calculated as:
$JV_{PSII}$ x $\varphi_{E:C}$ x $MW_C$ / Chl $a$
where $\varphi_{E:C}$ is the electron requirement for carbon uptake (molecule $CO_2$ (mol electrons)$^{-1}$), $MW_C$
is the molecular weight of carbon and Chl a is the Chl $a$ measurement specific to each sample.
Chl specific $JV_{PSII}$ based photosynthesis-irradiance curves were conducted in replicate batches
between 10:00 – 16:00 to account for variability over the photo-period at between 8 - 14
irradiance intensities. The maximum intensity applied was adjusted according to ambient
natural irradiance on the day of sampling. Maximum photosynthetic rates of carbon fixation
($P^B_m$), the light limited slope ($\alpha^B$) and the light saturation point of photosynthesis ($I_k$) were
estimated by fitting the data to the model by Webb et al., (1974):





$P^B = (1 - e \text{ x } (-\alpha \text{ x } I/P^B_m))$
Samples for FRRf fluorescence-based light curves were taken at T36.
**2.4 Statistical analysis**
To test for effects of high $pCO_2$, high temperature and high $pCO_2$ x high temperature on the
measured response variables (Chl *a*, total community biomass, POC, PON, photosynthetic
parameters and biomass of individual species), generalised least squares models with the
factors $pCO_2$, temperature and time (and all interactions) were applied to the data between T0
and T36 incorporating an auto-regressive correlation structure of the order (1) to account for
auto correlation. To test for significant differences between experimental treatments at T36 in
all measured parameters, generalized linear models were applied to the data. Where main
effects were established, pairwise comparisons were performed using the method of Herberich
*et al.*, (2010) for data with non-normality and/or heteroscedasticity. Weekly biomass values
from the L4 time-series were averaged over years to elucidate the variability and seasonal
cycles of the dominant species observed in the experimental community at T36, relative to the
time-series observations. The distribution of these species biomass at station L4 was also
analysed relative to the in-situ gradients of temperature (1993-2014) and $pCO_2$ (2008-2014)
using frequency histograms. Analyses were conducted using the R statistical package (R Core
Team (2014). R: A language and environment for statistical computing. R Foundation for
Statistical Computing, Vienna, Austria).
**3. Results**
Chl *a* concentration in the WEC ranged between 0.02-~5 mg m$^{-3}$ from 30 September - 6th
October 2015, with a concentration of ~1.6 mg m$^{-3}$ at station L4 (**Fig. 1 A**). Over the period
leading up to phytoplankton community sampling, increasing nitrate and silicate concentrations
coincided with a Chl *a* peak on 23rd September (**Fig. 1 B**). Routine net trawl (20 μm) sample
observations indicated a phytoplankton community dominated by the diatoms *Leptocylindrus*
*danicus* and *L. minimus* with a lower presence of the dinoflagellates *Prorocentrum cordatum*,
*Heterocapsa* spp. and *Oxytoxum gracile*. Following decreasing nitrate concentrations, this
community transitioned to a *P. cordatum* bloom on 29th September, the week before
experimental community sampling (data not shown).
**3.1 Experimental carbonate system**
Equilibration to the target high $pCO_2$ values (800 μatm) within the high $pCO_2$ and combination
treatments was achieved at T10 (**Fig. 2 A**). These treatments were slowly acclimated to
increasing levels of $pCO_2$ over 7 days (from the initial dilution at T3) while the control and high




temperature treatments were acclimated at the same ambient carbonate system values as that
from station L4 on the day of sampling. Following equilibration, the mean $pCO_2$ values within
the control and high temperature treatments were 394.9 (± 4.3 sd) and 393.2 (± 4.8 sd) µatm
respectively, while in the high $pCO_2$ and combination treatments mean $pCO_2$ values were 822.6
(± 9.4) and 836.5 (± 15.6 sd) µatm, respectively. Carbonate system values remained stable
throughout the experiment (**Fig. 2 B-D**).

### 3.2 Experimental temperature treatments

Mean temperatures in the control and high $pCO_2$ treatments were 14.1 (± 0.35 sd) °C and in the
high temperature and combination treatments the mean temperatures were 18.6 (± 0.42 sd) °C.
There was a mean temperature difference between the ambient and high temperature
treatments of 4.46 (± 0.42 sd) °C (Supporting information, Fig. S1 A & B).

### 3.3 Chlorophyll *a*

Mean Chl *a* in the experimental seawater at T0 was 1.64 (± 0.02 sd) mg m$^{-3}$ (**Fig. 3 A**). This
decreased in all treatments between T0 to T7, to ~0.1 (± 0.09, 0.035 and 0.035 sd) mg m$^{-3}$ in the
control, high $pCO_2$ and combination treatments, while in the high temperature treatment at T7
Chl *a* was 0.46 mg m$^{-3}$ (± 0.29 sd). From T7 to T12 there was an increase in Chl *a* in all
treatments which was highest in the combination (4.99 mg m$^{-3}$ ± 0.69 sd) and high $pCO_2$
treatments (3.83 mg m$^{-3}$ ± 0.43 sd) (**Table 1**). At T36 Chl *a* concentration in the combination
treatment was significantly higher than all other treatments at 6.87 (± 0.58 sd) mg m$^{-3}$ (**Table
2**) while the high temperature treatment concentration was significantly higher than the control
and high $pCO_2$ treatments at 4.77 (± 0.44 sd) mg m$^{-3}$ (**Table 2**). Mean concentrations for the
control and high $pCO_2$ treatments at T36 were not significantly different at 3.30 (± 0.22 sd) and
3.46 (± 0.35 sd) mg m$^{-3}$ respectively (pairwise comparison $t = 0.78$, $p = 0.858$).

### 3.4 Phytoplankton biomass

The starting biomass in all treatments was 110.2 (± 5.7 sd) mg C m$^{-3}$ (**Fig. 3 B**) and the
community biomass was dominated by dinoflagellates (~50%) with smaller contributions from
nanophytoplankton (~13%), cryptophytes (~11%), diatoms (~9%), coccolithophores (~8%),
S*ynechococcus* (~6%) and picophytoplankton (~3%). Total biomass increased significantly in
all treatments over time (**Table 1**) and at T10, it was significantly higher in the high
temperature treatment when the biomass reached 752 (± 106 sd) mg C m$^{-3}$. At T36 however,
total biomass was significantly higher in the high $pCO_2$ treatment (**Table 1**) and reached 2481



(± 182.68 sd) mg C m$^{-3}$, which increased more than 20-fold from T0. Total biomass in the high
temperature treatment increased more than 15-fold to 1735 (± 169.24 sd) mg C m$^{-3}$ at T36 and
was significantly higher than the combination treatment and ambient control, which were 525
(± 28.02 sd) mg C m$^{-3}$ and 378 (± 33.95 sd) mg C m$^{-3}$, respectively (**Table 2**).
Measured POC followed the same trends as estimated biomass in all treatments between T0 and
T36 (**Fig. 3 C**) and despite some variability between the two measures, POC was within the
range of estimates ($R^2$ = 0.914, **Fig. 3 D**).  At T36, POC was significantly greater in the high $pCO_2$
treatment (2086 ± 155.19 sd mg m$^{-3}$) followed by the high temperature treatment (1594 ±
162.24 sd mg m$^{-3}$), which were significantly greater than the control and combination treatment
(**Table 1**). PON followed the same trends as POC over the course of the experiment (**Fig. 3 E,**
**Table 1**): at T36 concentrations were 147 (± 12.99 sd) and 133 (± 15.59 sd) mg m$^{-3}$ in the high
$pCO_2$ and high temperature treatments respectively, while PON was 57.75 (± 13.07 sd) mg m$^{-3}$
in the combination treatment and 47.18 (± 9.32 sd) mg m$^{-3}$ in the control (**Table 1**). POC:PON
ratios increased significantly over time in all treatments except for the control. The largest
increase of 33 %, from 10.72 to 14.26 mg m$^{-3}$ (± 1.73 sd) was in the high $pCO_2$ treatment,
followed by an increase of 32 % to 9.83 (± 1.82 sd) mg m$^{-3}$ in the combination treatment, and an
increase of 17 % to 12.09 (± 2.14 sd) mg m$^{-3}$ in the high temperature treatment.  In contrast, the
POC:PON ratio in the control declined by 20 % from T0 to T36, from 10.33 to 8.26 (± 0.50 sd)
mg m$^{-3}$ (**Fig. 3 F, Table 1**).

### 3.5 Community composition

At T36 diatoms dominated the phytoplankton community biomass in the ambient control with a
substantial contribution from nanophytoplankton (**Fig. 4 A**), while the high temperature and
high $pCO_2$ treatments exhibited near mono-specific dominance of nanophytoplankton (**Figs. 4 B**
**& C**). The most diverse community was in the combination treatment where dinoflagellates and
*Synechococcus* became more prominent (**Fig. 4 D**).
Between T10 and T24 the community shifted to nanophytoplankton in all experimental
treatments. This dominance was maintained through to T36 in the high temperature and high
$pCO_2$ treatments whereas in the ambient control and combination treatment, the community
shifted away from nanophytoplankton (**Fig. 5 A**). At T36 nanophytoplankton biomass was
significantly higher in the high $pCO_2$ treatment followed by the high temperature treatment
(**Table 2**) when biomass attained 2216 (± 189.67 sd) mg C m$^{-3}$ and 1489 (± 170.32 sd) mg C m$^{-3}$,
respectively. In the combination treatment nanophytoplankton biomass was 238 (± 14.16 sd)
mg C m$^{-3}$ at T36 which was significantly higher compared to the ambient control (162 ± 20.02 sd
mg C m$^{-3}$; **Table 2**). In addition to significant differences in nanophytoplankton biomass



amongst the experimental treatments, treatment-specific differences in cell size was observed.
Larger nano-flagellates dominated the control (mean cell diameter of 6.34 μm), smaller nano-
flagellates dominated the high temperature and combination treatments (mean cell diameters
of 3.61 μm and 4.28 μm) whereas *Phaeocystis* spp. dominated the high $pCO_2$ treatment (mean
cell diameter 5.04 μm) and was not observed in any other treatment (Supporting Information,
Fig. S2 A-D).
Low starting biomass of diatoms at T0 was dominated by *Coscinodiscus wailessi* (48 %; 4.99 mg
C m$^{-3}$), *Pleurosigma* (25 %; 2.56 mg C m$^{-3}$) and *Thalassiosira subtilis* (19 %; 1.94 mg C m$^{-3}$). Small
biomass contributions were made by *Navicula distans*, undetermined pennate diatoms and
*Cylindrotheca closterium*. Biomass in the diatom group remained low from T0 to T24 but
increased at T36 in all treatments, with significantly higher biomass in the high $pCO_2$ treatment
(235 ± 21.41 sd mg C m$^{-3}$, **Fig. 5 B, Table 2**). The highest diatom contribution to total
community biomass at T36 was in the ambient control (52 % of biomass; 198 ± 17.28 sd mg C
m$^{-3}$). In both the high temperature and combination treatments diatom biomass was
significantly lower at T36 (151 ± 10.94 sd and 124 ± 19.16 sd mg C m$^{-3}$, respectively). In all
treatments at T36, diatom biomass shifted away from dominance of the larger *C. Wailessii* to the
comparatively smaller *C. closterium*, *N. distans*, *T. subtilis* and *Tropidoneis* spp., the relative
contributions of which were treatment-specific. Overall *N. distans* dominated diatom biomass in
all treatments at T36 (ambient control: 112 ± 24.86 sd mg C m$^{-3}$, 56 % of biomass; high
temperature: 106 ± 17.75 sd mg C m$^{-3}$, 70 % of biomass; high $pCO_2$: 152 ± 19.09 sd mg C m$^{-3}$, 61
% of biomass; and combination: 111 ± 20.97 sd mg C m$^{-3}$, 89 % of biomass; Supporting
Information, Fig. S3 A-D).
The starting dinoflagellate community was dominated by *Gyrodinium spirale* (91 %; 49 mg C m$^{-3}$
), with smaller contributions from *Katodinium glaucum* (5 %; 2.76mg C m$^{-3}$), *Prorocentrum*
*cordatum* (3 %; 1.78 mg C m$^{-3}$) and undetermined *Gymnodiniales* (1 %; 0.49 mg C m-3). At T36
dinoflagellate biomass was significantly higher in the combination treatment (90 ± 16.98 sd mg
C m$^{-3}$, **Fig. 5 C, Table 2**) followed by the high temperature treatment (57 ± 6.87 sd mg C m$^{-3}$,
**Table 2**). There was no significant difference in dinoflagellate biomass between the high $pCO_2$
treatment and ambient control at T36 when biomass was low. In the combination treatment,
dinoflagellate biomass shifted away from the larger *G. spirale* and was dominated by *P.*
*cordatum* which contributed 59 ± 12.95 sd mg C m$^{-3}$ (66 % of biomass in this group).
*Synechococcus* biomass was significantly higher at T36 in the combination treatment (59.9 ±
4.30 sd mg C m$^{-3}$, **Fig. 5 D, Table 2**) followed by the high temperature treatment (30 ± 5.98 sd
mg C m$^{-3}$, **Table 2**). In both the high $pCO_2$ treatment and ambient control at T36 *Synechococcus*
biomass was low (~7 mg C m$^{-3}$ in both treatments). Relative to the other phytoplankton groups,





biomass of picophytoplankton (**Fig. 5 E**), cryptophytes (**Fig. 5 F**) and coccolithophores (**Fig. 5**
**G**) remained low in all treatments throughout the experiment. Though picophytoplankton
responded positively to the high $pCO_2$ and combination treatments at T36 (high $pCO_2$: 6.93 ±
0.63 sd mg C m$^{-3}$; combination: 11.26 ± 0.79 sd mg C m$^{-3}$; **Table 2**).

### 3.6 Chl fluorescence-based photophysiology

At T36, FRRf PI parameters were strongly influenced by the experimental treatments. $P^B_m$ was
significantly higher in the high $pCO_2$ treatment (18.93 mg C (mg Chl $a$)$^{-1}$ m$^{-3}$ h$^{-1}$), followed by the
high temperature treatment (9.58 mg C (mg Chl $a$)$^{-1}$ m$^{-3}$ h$^{-1}$; **Fig. 6, Tables 3 & 4**). There was no
significant difference in $P^B_m$ between the ambient control and combination treatment (2.77 and
3.02 mg C (mg Chl $a$)$^{-1}$ m$^{-3}$ h$^{-1}$). Light limited photosynthetic efficiency ($\alpha^B$) also followed the
same trend and was significantly higher in the high $pCO_2$ treatment (0.13 mg C (mg Chl $a$)$^{-1}$ m$^{-3}$
h$^{-1}$ (μmol photon m$^{-2}$ s$^{-1}$)$^{-1}$) followed by the high temperature treatment (0.09 mg C (mg Chl $a$)$^{-1}$
m$^{-3}$ h$^{-1}$ (μmol photon m$^{-2}$ s$^{-1}$)$^{-1}$) (**Tables 3 & 4**). $\alpha^B$ was low in both control and combination
treatments (0.03 and 0.04 mg C (mg Chl $a$)$^{-1}$ m$^{-3}$ h$^{-1}$ (μmol photon m$^{-2}$ s$^{-1}$)$^{-1}$, respectively). The
light saturation point of photosynthesis ($I_k$) was significantly higher in the high $pCO_2$ treatment
relative to all treatments where $I_k$ was 144.13 μmol photon m$^{-2}$ s$^{-1}$, though significantly lower in
the combination treatment relative to both the high $pCO_2$ and high temperature treatments
(**Tables 3 & 4**).

### 3.7 Natural variability of biomass in the WEC, station L4 time series.

Nanophytoplankton is a critical component of the station L4 carbon budget. The mean annual
total nanophytoplankton biomass over the time series (1993-2014) was 586 (± 16.54 sd) mg C
m$^{-3}$ with maximum annual biomass of 1182 mg C m$^{-3}$ in 2002 (62 % of total annual
phytoplankton biomass) and minimum annual biomass of 262 mg C m$^{-3}$ in 2008 (23 % of total
annual phytoplankton biomass). In 8 of the 21 years of time series observations,
nanophytoplankton contributed more than 40 % of the station L4 carbon budget. Consistently
over the seasonal cycle at L4, mean nanophytoplankton biomass > 10 mg C m$^{-3}$ occurred from
early April until the end of October, exhibiting sustained long-term seasonality relative to other
phytoplankton groups, though maximal biomass was constrained between April and the 3$^{rd}$
week of June with one exception (**Fig. 7 A**).
*N. distans* dominated diatom biomass in the experimental communities though was found to be
a very minor component of the diatom carbon budget at station L4 (0.04 % of total annual
diatom biomass). Weekly *N. distans* biomass averaged over the time series was very low and
ranged from below the limit of detection to ~0.2 mg C m$^{-3}$ with maximum total annual biomass

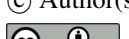



of ~0.5 mg C m⁻³ in 2005 (**Fig. 7 B**). Seasonality of maximal *N. distans* biomass was constrained
to September-October when the mean maximal biomass was 0.03 mg C m⁻³.
*P. cordatum* dominated the dinoflagellate biomass in the experimental communities, and made a
significant contribution to total biomass in the combination treatment. Weekly *P. cordatum*
biomass averaged over the time series at L4 ranged from 0.004 to 107 mg C m⁻³ and exhibited
strong seasonality. Mean total annual *P. cordatum* biomass was 25.5 mg C m⁻³ with maximum
annual biomass of 233 mg C m⁻³ in 2006 (minimum annual biomass of 0.004 mg C m⁻³ in 1994).
The bloom peak (taken as an increase in biomass > 1.0 mg C m⁻³) usually occurred in September
although as early as mid-June (2001 and 2013) in some cases (**Fig. 7 C**). Mean maximal biomass
was 12.7 mg C m⁻³ with positive anomalies occurring in 5 out of 21 years throughout the time-
series (ranging from 15 to 107 mg C m⁻³). *P. cordatum* contributed on average, 9.2 % of the total
annual dinoflagellate biomass with a maximum contribution of ~55 % in 2006 and 12 % and 63
% when averaged over the bloom period from mid-June to end-September. *P. cordatum*
contributed 3.4 % of total phytoplankton biomass during the bloom period and ~32 % of total
phytoplankton biomass in 2006 during an unprecedented bloom when biomass attained 107
mg C m⁻³ in 2006 (**Fig. 7 D**).
Group and species-specific optimal temperature ranges were found when considering how the
dominant experimental species were temporally distributed relative to in-situ temperatures:
nanophytoplankton exhibited a bi-modal distribution with 31 % biomass between 9-11 °C and
30 % between 15-16.5 °C, with 6 % above 16.5 °C (**Fig. 8 A**). ~60 % of *N. distans* biomass
occurred between 14-16 °C and 2 % biomass above 16 °C (**Fig. 8 B**). In contrast, 66 % of *P.*
*cordatum* biomass occurred between 14-16 °C and 24 % above 16 °C (**Fig. 8 C**). Biomass
distribution relative to station L4 in-situ $pCO_2$ levels (2008-2014) also demonstrated
group/species-specific optimal ranges. 71 % nanophytoplankton biomass occurred at a $pCO_2$
range of 245-410 µatm, 24 % 410-515 µatm and 5 % between 515-680 µatm (**Fig. 8 D**). *N.*
*distans* followed a similar trend with 72% biomass between a $pCO_2$ range of 245-410 µatm, 26%
between 410-515 µatm and < 2% beyond 515 µatm (**Fig. 8 E**). By contrast, 97% of *P. cordatum*
biomass occurred between 245-410 µatm with 3 % biomass occurring beyond 410 µatm (**Fig. 8**
**F**).
**4. Discussion**
Individually, elevated temperature and $pCO_2$ resulted in the highest biomass and maximum
photosynthetic rates ($P^B_m$), when nanophytoplankton dominated. The interaction of these two
factors had little effect on total biomass with values close to the ambient control, and no effect





on $P^B_m$. The combination treatment, however, exhibited the greatest diversity of phytoplankton
functional groups with dinoflagellates and *Synechococcus* becoming more prominent.
Elevated $pCO_2$ has been shown to enhance the growth and photosynthesis of some
phytoplankton species which have active uptake systems for inorganic carbon (Giordano et al.,
2005; Reinfelder, 2011). Elevated $pCO_2$ may therefore lead to lowered energetic costs of carbon
assimilation in some species and a redistribution of the cellular energy budget to other
processes (Tortell et al., 2002). In the present study, under elevated $pCO_2$ where the dominant
group was nanophytoplankton, the community was dominated by the bloom-forming
haptophyte *Phaeocystis* spp. Photosynthetic carbon fixation in *Phaeocystis* spp. is presently near
saturation with respect to current levels of $pCO_2$ (Rost et al., 2003). Inorganic carbon acquisition
in this species has been shown to be equal to, or more efficient than that of diatoms. Indeed, in
*Phaeocystis* spp, extracellular carbonic anhydrase is regulated by $CO_2$ (aq), and $HCO_3^-$ is utilized
as a carbon source in photosynthesis, indicating more efficient use of $CO_2$ (Elzenga et al., 2000;
Rost et al., 2003), and thus providing an advantage to *Phaeocystis* spp. when more $CO_2$ is
present. Therefore, the increased biomass and photosynthetic carbon fixation seen here under
elevated $pCO_2$ can be attributed to the community shift to *Phaeocystis* spp.. The  increased
biomass seen in the high temperature treatment in this study, may be attributed to enhanced
enzymatic activities, since algal growth commonly increases with temperature until after the
optimal range (Boyd et al., 2013; Goldman and Carpenter, 1974; Savage et al., 2004) and
optimum growth temperatures for marine phytoplankton are often several degrees higher than
environmental temperatures (Eppley, 1972; Thomas et al., 2012).
**4.1 Chl *a***
Chl *a* concentration was significantly higher in the combination treatment at T36 when total
biomass was lower, but Chl *a* was significantly lower in the high $pCO_2$ treatment when biomass
was significantly higher than all other treatments. This contrasts the results reported in
comparable studies as Chl *a* is generally highly correlated with biomass, ( e.g. Feng et al., 2009).
Similar results were reported however by Hare et al., (2007) which indicates that Chl *a* may not
always be a reliable proxy for biomass in mixed communities. Differences in Chl *a* may therefore
be attributed to taxonomic differences in community composition.
**4.2 Biomass**
This study shows that the phytoplankton community response to elevated temperature and
$pCO_2$ is highly variable. $pCO_2$ elevated to ~800 µatm induced higher community biomass in
agreement with Kim et al., (2006) and Riebesell et al., (2007), whereas in other natural
community studies no $CO_2$ effect on biomass was observed (Delille et al., 2005; Maugendre et al.,



2017; Paul et al., 2015). A ~4.5 °C increase in temperature also resulted in higher biomass in
this study, similar to the findings of Feng et al., (2009) and Hare et al., (2007) though elevated
temperature has previously reduced biomass of natural nanophytoplankton communities in the
Western Baltic Sea and Arctic Ocean (Coello-Camba et al., 2014; Moustaka-Gouni et al., 2016).
When elevated temperature and $pCO_2$ were combined, community biomass exhibited little
response, similar to the findings of Gao et al., (2017), though an increase in biomass has also
been reported (Calbet et al., 2014; Feng et al., 2009).  Geographic location and season also play
an important role in structuring the community and its response in terms of biomass to elevated
temperature and $pCO_2$, e.g. (Li et al., 2009; Morán et al., 2010).

### 477    4.3 Carbon:Nitrogen

In agreement with others, the results of this experiment showed highest increases in C:N under
elevated $pCO_2$ alone (Riebesell et al., 2007). C:N also increased under high temperature,
consistent with the findings of Lomas and Glibert, (1999) and Taucher et al., (2015) and was
stimulated to a lesser degree when $pCO_2$ and temperature were elevated simultaneously, which
was also observed in the study of Calbet et al., (2014), but contrasts other studies that have
observed C:N to be unaffected by the combined influence of elevated $pCO_2$ and temperature, e.g.
(Deppeler and Davidson, 2017; Kim et al., 2006; C. Paul et al., 2015). C:N is a strong indicator of
cellular protein content (Woods and Harrison, 2003) and increases under elevated $pCO_2$ and
warming may likely lead to lowered nutritional value of phytoplankton with consequences for
zooplankton reproduction and biogeochemical cycles.

### 488    4.4 Photosynthetic carbon fixation rates

At T36, under elevated $pCO_2$ $P^B_m$ was > 6 times higher than the ambient control, which has also
been reported in elevated $pCO_2$ perturbation experiments by Riebesell et al., (2007) and Tortell
et al., (2008), but contrasts other observations on natural populations where the effect of
elevated $pCO_2$ alone was found to reduce $P^B_m$ (Feng et al., 2009; Hare et al., 2007). Studies on
laboratory cultures have shown that increases in temperature increase photosynthetic rates
(Feng et al., 2008; Fu et al., 2007; Hutchins et al., 2007), similar to our findings. We found that
there was no effect on $P^B_m$ under the combined treatment which has also been observed in
experiments on natural populations (Coello-Camba and Agustí, 2016; Gao et al., 2017). This
strongly contrasts the findings of Feng et al., (2009) and Hare et al., (2007) who observed the
highest $P^B_m$ when temperature and $pCO_2$ were elevated simultaneously. Increases in $\alpha^B$ and $I_k$
under elevated $pCO_2$, and a decrease in these parameters when elevated $pCO_2$ and temperature
were combined is opposite to the trends reported by Feng et al., (2009).





Photosynthetic rates have been demonstrated to decrease beyond a temperature of 20 °C
(Raven and Geider, 1988) which can be modified through photoprotective rather than
photosynthetic pigments (Kiefer and Mitchell, 1983). This may explain the difference in $P^B_m$
between the high $pCO_2$ and high temperature treatments (in addition to differences in
nanophytoplankton community composition in relation to *Phaeocystis* spp. discussed above), as
the experimental high temperature treatment in the present study was ~4.5 ° C higher than
ambient.
There was no significant effect of combined elevated $pCO_2$ and temperature on $P^B_m$, which was
strongly influenced by taxonomic differences between the experimental treatments. Warming
has been shown to lead to smaller cell sizes in nanophytoplankton (Atkinson et al., 2003; Peter
and Sommer, 2012), which was observed in the combined treatment together with decreased
nanophytoplankton biomass. Diatoms also shifted to smaller species with reduced biomass,
while dinoflagellate and *Synechococcus* biomass increased at T36. Dinoflagellates are the only
photoautotrophs with form II RuBisCO (Morse et al., 1995) which has the lowest
carboxylation:oxygenation specificity factor among eukaryotic phytoplankton (Badger et al.,
1998), giving dinoflagellates a disadvantage in carbon fixation under present ambient $pCO_2$
levels. Dinoflagellates generally grow at slower rates in surface waters with high pH (≥9)
resulting from photosynthetic removal of $CO_2$ by previous blooms (Hansen, 2002; Hinga, 2002).
Though growth under high pH provides indirect evidence that dinoflagellates possess CCMs,
direct evidence is limited and points to the efficiency of CCMs in dinoflagellates as moderate in
comparison to diatoms and some haptophytes (Reinfelder, 2011 and references therein). This
may explain the lower $P^B_m$ in the combined treatment compared to elevated $pCO_2$ and
temperature individually.
**4.5 Community composition**
Phytoplankton community structure changes were observed, with a shift from dinoflagellates to
nanophytoplankton which was most pronounced under single treatments of elevated
temperature and $pCO_2$. Amongst the nanophytoplankton, a distinct size shift to smaller cells was
observed in the high temperature and combination treatments, while in the high $pCO_2$
treatment *Phaeocystis* spp. dominated. Under combined $pCO_2$ and temperature at T36 however,
dinoflagellate and *Synechococcus* biomass increased at the expense of nanophytoplankton.
An increase in pico- and nanophytoplankton has previously been reported in natural
communities under elevated $pCO_2$ (Bermúdez et al., 2016; Boras et al., 2016; Brussaard et al.,
2013; Engel et al., 2008) while no effect on these size classes has been observed in other studies
(Calbet et al., 2014; Paulino et al., 2007). Moustaka-Gouni et al., (2016) also found no $CO_2$ effect



on natural nanophytoplankton communities but increased temperature reduced the biomass of
this group. Kim et al., (2006) observed a shift from nanophytoplankton to diatoms under
elevated $pCO_2$ alone while a shift from diatoms to nanophytoplankton under combined elevated
$pCO_2$ and temperature has been reported (Hare et al., 2007). A variable response in *Phaeocystis*
spp. to elevated $pCO_2$ has also been reported with increased growth (Chen et al., 2014; Keys et
al., 2017), no effect (Thoisen et al., 2015) and decreased growth (Hoogstraten et al., 2012)
observed. *Phaeocystis* spp. can outcompete other phytoplankton and form massive blooms (up
to 10 g C m$^{-3}$) with impacts on food webs, global biogeochemical cycles and climate regulation
(Schoemann et al., 2005). While not a highly toxic algal species, *Phaeocystis* spp. are considered
a harmful algal bloom (HAB) species when biomass reaches sufficient concentrations to cause
anoxia through the production of mucus foam which can clog the feeding apparatus of
zooplankton and fish (Eilertsen & Raa, 1995).
The response of diatoms to elevated $pCO_2$ and temperature has been variable. For example, A
study by Taucher et al., (2015) showed that *Thalassiosira weissflogii* incubated at 1000 µatm
$pCO_2$ increased growth by 8 % while for *Dactyliosolen fragilissimus*, growth increased by 39 %;
temperature elevated by + 5°C also had a stimulating effect on *T. weissflogii* but inhibited the
growth rate of *D. fragilissimus*; and when the treatments were combined growth was enhanced
in *T. weissflogii* but reduced in *D. fragilissimus*. In partial agreement, the results of the present
experiment show that elevated $pCO_2$ increased biomass in diatoms but elevated temperature
and the combination of these factors reduced biomass. A distinct size-shift in diatom species
was observed in all treatments, from the larger *Coscinodiscus* spp., *Pleurosigma* and
*Thalassiosira subtilis* to the smaller *Navicula distans*. This was most pronounced in the
combination treatment where *N. distans* contributed 89 % of diatom biomass. *Navicula* spp.
previously exhibited a differential response to both elevated temperature and $pCO_2$. At + 4.5 °C
and 960 ppm $CO_2$ Torstensson et al., (2012) observed no synergistic effects on the benthic
*Navicula directa*. Elevated temperature increased growth rates by 43 % while a reduction of 5 %
was observed under elevated $CO_2$. No effects on growth were detected at pH ranging from 8 –
7.4 units on *Navicula* spp. (Thoisen et al., 2015), while growth in *N. distans* was significantly
stimulated along a $CO_2$ gradient at a shallow cold-water vent system (Baragi et al., 2015).
*Synechococcus* grown under $pCO_2$ elevated to 750 ppm and temperature elevated by 4 °C
resulted in increased growth and a 4-fold increase in $P^B_m$ (Fu et al., 2007) which is similar to the
results of the present study.
The combination of elevated temperature and $pCO_2$ significantly increased dinoflagellate
biomass which almost doubled, accounting for 17 % of total biomass. This was due to *P.*
*cordatum* which increased biomass by more than 30-fold between T0 and T30 (66 % of



dinoflagellate biomass in this treatment). Despite the global increase in the frequency of HABs
few studies have focussed on the response of dinoflagellates to elevated $pCO_2$ and temperature.
In laboratory studies at 1000 ppm $CO_2$, growth rates of the HAB species *Karenia brevis* increased
by 46 %, at 1000 ppm $CO_2$ and + 5 °C temperature it's growth increased by 30 % but was
reduced under elevated temperature alone (Errera et al., 2014).  A combined increase in $pCO_2$
and temperature enhanced both the growth and $P^B_m$ in the dinoflagellate *Heterosigma akashiwo*,
whereas in contrast to the present findings, only $pCO_2$ alone enhanced these parameters in *P.*
*cordatum* (Fu et al., 2008).
Among HAB species, *P. cordatum* is widely distributed geographically in temperate and
subtropical waters, has detrimental effects at the organismal and environmental levels and is
potentially harmful to humans via shellfish poisoning (Heil et al., 2005). Recent increases in the
frequency, magnitude and distribution of harmful phytoplankton species has focussed attention
on the unique physiological, ecological and toxicological aspects of the species involved
(Andeson et al., 2002; Hallegraeff, 1993). Ocean acidification combined with warming could
potentially affect the growth and toxicity of HAB species (Fu et al., 2012). Recent studies on
several diatom and dinoflagellate species suggest that ocean acidification combined with
elevated temperature may dramatically increase the toxicity of some harmful groups (e.g.
Flores-Moya et al., 2012; Fu et al., 2010; Sun et al., 2011; Tatters et al., 2012). The ecology and
bloom dynamics of *P. cordatum* have been well documented in selected environments (e.g.
Chesapeake Bay, Baltic Sea). The spread of this species to previously unreported areas through
either ballast water transport, aquaculture development, or increasing eutrophication, has been
reported (Heil et al., 2005). In Chesapeake Bay *P. cordatum* 'mahogany tides' have been
associated with anoxic/hypoxic events, fish kills, aquaculture shellfish kills and the loss of
aquatic vegetation (Tango et al., 2005). Over the last two decades *P. cordatum* has established
itself as a dominant summer phytoplankton species in the Baltic Sea but so far there are no
reports of toxic blooms (Hajdu et al., 2000). However, for the first time a *P. cordatum* bloom was
recorded in February 2002 at Bolinao, Northern Philippines and was coincident with a mass
aquaculture fish kill resulting in losses estimated at US$120,000 (Azanza et al., 2005). Several
clones of *P. cordatum* were found to produce a water-soluble neuro-toxin during bloom decline
in culture studies (Grzebyk et al., 1997). More recently, a series of positive bioassays for
tetrodotoxins (TTXs) was observed in mussels (Vlamis et al., 2015) which coincided with the
simultaneous presence of a *P. cordatum* bloom. Data analysis from previous years (2006 –
2012) identified multiple sample cases for toxins in aquaculture production areas coinciding
with *P. cordatum* blooms.

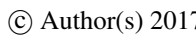


In addition to strong links to toxic algal events, mixotrophy has also been reported in *P.*
*cordatum*. In a study by Stoecker et al., (1997) up to 50% of *P. cordatum* sampled from
Chesapeake Bay in the summer contained cryptophyte material. The authors concluded that *P.*
*cordatum* feeding is a mechanism for supplementing carbon nutrition and this may explain why
the ratio of nanophytoplankton:dinoflagellates was significantly lower in our combination
experimental community compared to the other treatments.
**4.6 Natural variability of biomass in the WEC, station L4 time series.**
During autumn in the WEC, sea surface temperature and $pCO_2$ start to decline following their
respective time series maximal values at station L4. During October, mean seawater
temperatures at 10 m decrease from 15.39 °C (± 0.49 sd) to 14.37 °C (± 0.62 sd). Following a
period of $CO_2$ oversaturation in late summer, $pCO_2$ returns to near-equilibrium at station L4 in
October when mean $pCO_2$ values decrease during this month from 455.32 µatm (± 63.92 sd) to
404.06 µatm (± 38.55 sd) (Kitidis et al., 2012). As is the case with seawater warming, predicted
future ocean acidification is likely to impact coastal phytoplankton communities in autumn
when the present upper limit of the $pCO_2$ threshold increases during this period of surface
ocean-atmosphere equilibrium (Riebesell, 2004).
From a biological perspective, the autumn period at station L4 is characterised by the decline of
the late summer diatom and dinoflagellate blooms (Widdicombe et al., 2010) when biomass of
these two groups approaches values close to the time series minima (diatom biomass range:
6.01 (± 6.88 sd) – 2.85 (± 3.28 sd) mg C m$^{-3}$; dinoflagellate biomass range: 1.75 (± 3.28 sd) – 0.66
(± 1.08 sd) mg C m$^{-3}$). Typically, over this period nanophytoplankton becomes numerically
dominant when biomass of this group ranges from 20.94 (± 33.25 sd) – 9.38 (± 3.31 sd), though
the time series shows high variability in this biomass.
Comparative analyses of the WEC time series and the dominant species from the experimental
treatments showed that nanophytoplankton contributes significantly to the station L4 carbon
budget. The in-situ bimodal distribution of nanophytoplankton biomass at cold and warm
temperature ranges indicates a potential tolerance to temperature increase. Most
nanophytoplankton biomass occurred at an in-situ $pCO_2$ range between 245-410 µatm but
almost 10 % of biomass was distributed between 515-680 µatm, indicating some tolerance to
elevated $pCO_2$ during periods of $CO_2$ oversaturation at station L4. The dominant diatom species
in the experimental communities, *N. distans*, was a very minor contributor to diatom biomass
over the time series with most biomass constrained to very narrow in-situ temperature and
$pCO_2$ ranges of 14-16 ° C and 245-410 µatm. *P. cordatum* dominated dinoflagellate biomass in
the combination treatment but was generally a low biomass contributor to dinoflagellate





biomass over the time series, with the exception of one unprecedented bloom in 2006. *P.*
*cordatum* biomass exhibited higher thermal in-situ optima with most biomass observed
between 14-16 ° C and almost a quarter of biomass above 16 °C, indicating tolerance to
temperature increase, though the majority of biomass at station L4 occurred at times of low in-
situ $pCO_2$ (245-350 µatm) with just 3% beyond 410 µatm. These trends suggest conditions of
warming may favour nanophytoplankton and *P. cordatum*, elevated $pCO_2$ may favour
nanophytoplankton and both factors combined may favour both species. These observations are
consistent with the experimental results.

### 646    5.  Implications

Increased biomass, $P^B_m$ and a community shift to nanophytoplankton under individual increases
in temperature and $pCO_2$ suggests a potential positive feedback on atmospheric $CO_2$, whereby
more $CO_2$ is removed from the ocean, and hence from the atmosphere by photosynthetic
activity. The selection of *Phaeocystis* spp. under elevated $pCO_2$ indicates the potential for
negative impacts on ecosystem function and food web structure associated with this species
(Schoemann et al., 2005; Verity et al., 2007). However, while more $CO_2$ is photosynthesised,
selection for nanophytoplankton in both of these treatments may actually result in reduced
carbon sequestration due to slower sinking rates of these smaller phytoplankton cells (Bopp et
al., 2001; Laws et al., 2000). When temperature and $pCO_2$ were elevated simultaneously,
community biomass showed little response and no effects on $P^B_m$ were observed, suggesting a
negative feedback on atmospheric $CO_2$ and climate warming in future warmer high $CO_2$ oceans.
Additionally, combined elevated $pCO_2$ and temperature significantly modified taxonomic
composition, by reducing diatom biomass relative to the ambient control with increasing
dinoflagellate biomass dominated by the HAB species, *P. cordatum*. This has implications for
fisheries, ecosystem function and human health.

### 662    6.  Conclusion

These experimental results provide new evidence that increases in $pCO_2$ coupled with rising sea
temperatures may have antagonistic effects on the autumn phytoplankton community in the
WEC. Under future global change scenarios, the size range and biomass of diatoms may be
reduced with increased dinoflagellate biomass and the selection of HAB species. The
experimental simulations of year 2100 temperature and $pCO_2$ demonstrate that the effects of
warming can be offset by elevated $pCO_2$ potentially reducing coastal phytoplankton productivity
and significantly altering the community structure, and in turn these shifts will have
consequences on carbon biogeochemical cycling in the WEC.





*Data availability*: Experimental data used for analysis will be made available (DOI will be created)

*Author contributions*: Matthew Keys collected, measured, processed and analysed the data and prepared the figures. Drs Gavin Tilstone and Helen Findlay conceived, directed and sought the necessary funds to support the research. Matthew Keys and Dr Gavin Tilstone wrote the paper with input from Claire Widdicombe and Professor Tracy Lawson. Claire Widdicombe supervised and advised on phytoplankton taxonomic classifications.

*Competing interests*: The authors declare that they have no conflict of interest.

*Acknowledgements*: G.H.T, H.S.F. and C.E.W were supported by the UK Natural Environment Research Council's (NERC) National Capability – The Western English Channel Observatory (WCO). C.E.W was also partly funded by the NERC and Department for Environment, Food and Rural Affairs, Marine Ecosystems Research Program (Grant no. NE/L003279/1). M.K. was supported by a NERC PhD studentship (grant No. NE/L50189X/1). We thank Glen Tarran for his training, help and assistance with flow cytometry, The National Earth Observation Data Archive and Analysis Service UK (NEODAAS) for their help in providing the MODIS image used in Fig 1. and the crew of RV Plymouth Quest for their helpful assistance during field sampling.

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



.015

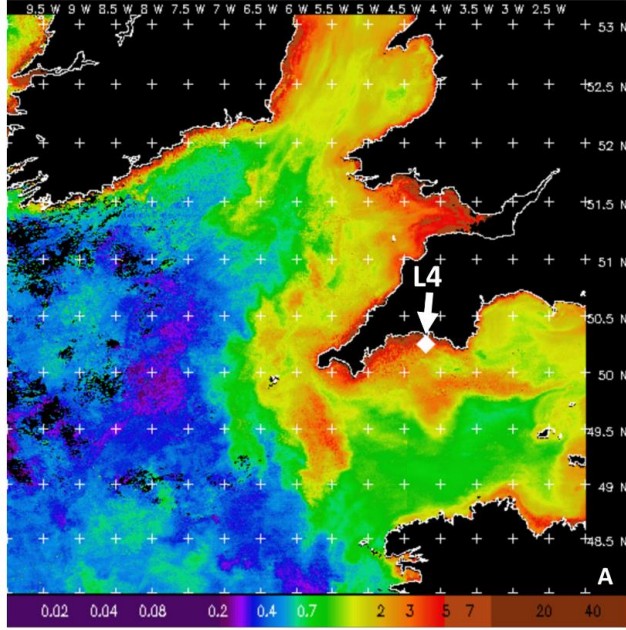

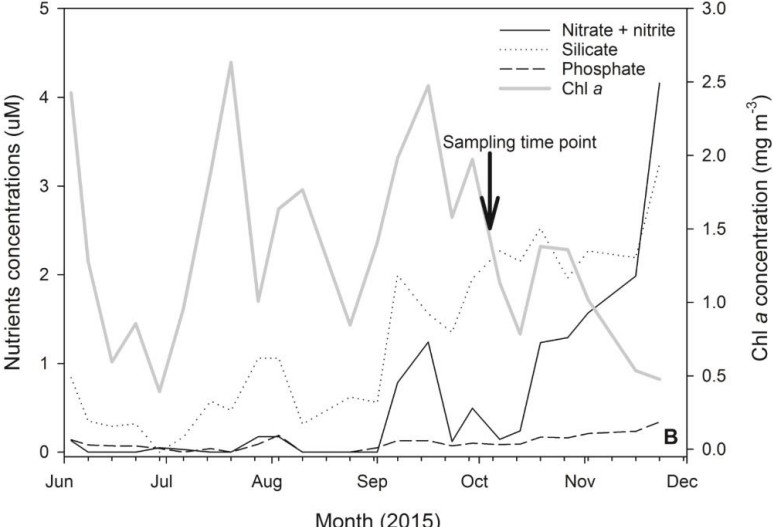

**Fig. 1. (A).** MODIS weekly composite chl *a* image of the western English Channel covering the period 30th September – 6th October 2015 (coincident with the week of phytoplankton community sampling for the present study), processing courtesy of NEODAAS. The position of coastal station L4 is marked with a white diamond. **(B).** Profiles of weekly nutrient and chl *a* concentrations from station L4 at a depth of 10 m over the second half of 2015 in the months prior to phytoplankton community sampling (indicated by black arrow and text).

.016





**Fig. 2.** Carbonate system values of the experimental phytoplankton incubations. **(A).** partial pressure of $CO_2$ in seawater ($pCO_2$), **(B).** pH on the NBS scale, **(C).** carbonate concentration ($CO_3^{2-}$) and **(D).** bicarbonate concentration ($HCO_3^-$) were estimated from direct measurements of total alkalinity and dissolved inorganic carbon.









**Fig. 3.** Time course of chl *a* (**A**), estimated phytoplankton biomass (**B**), POC (**C**), regression of estimated phytoplankton carbon vs measured POC (**D**), PON (**E**) and POC:PON (**F**).



.023

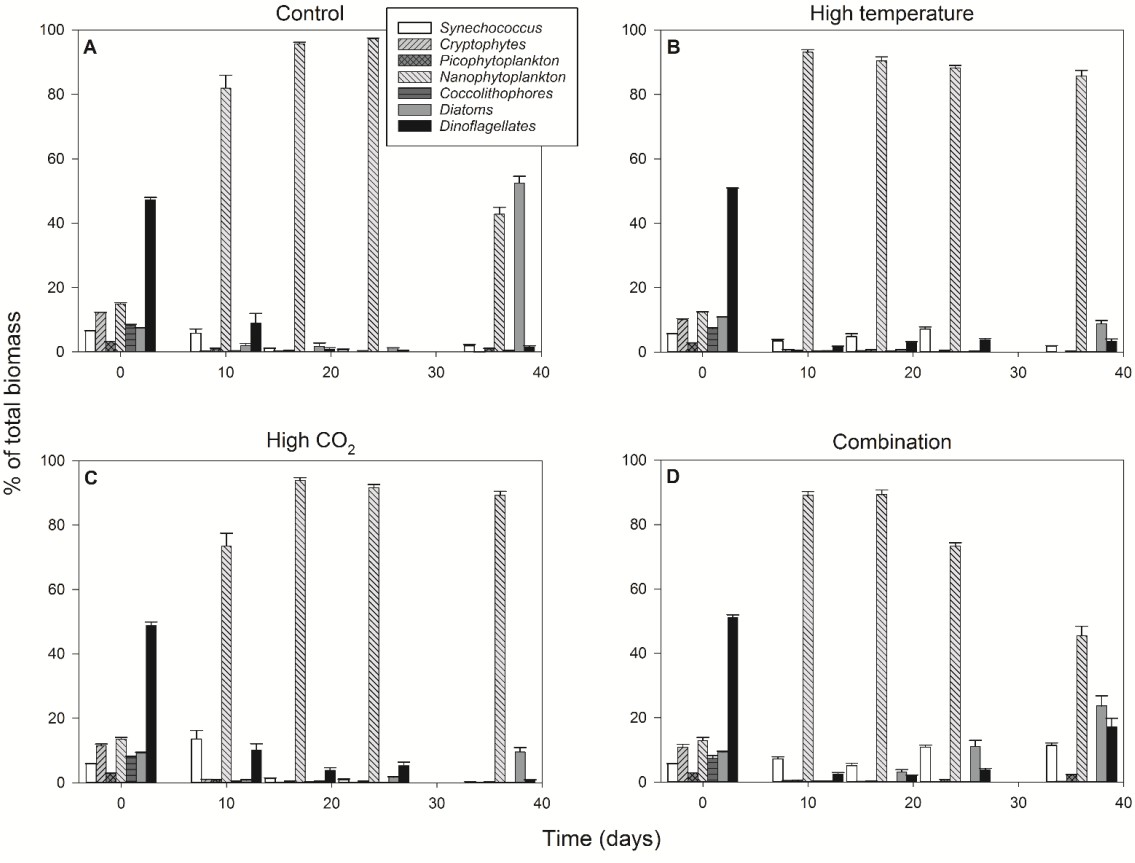

**Fig. 4.** Percentage contribution to community biomass by phytoplankton groups/species throughout the experiment in the control (**A**), high temperature (**B**), high $CO_2$ (**C**) and combination treatments (**D**).

.024

.025

.026

.027

.028

.029

.030

.031

.032

.033

.034

.035



**Fig. 5.** Response of individual phytoplankton groups to experimental treatments.



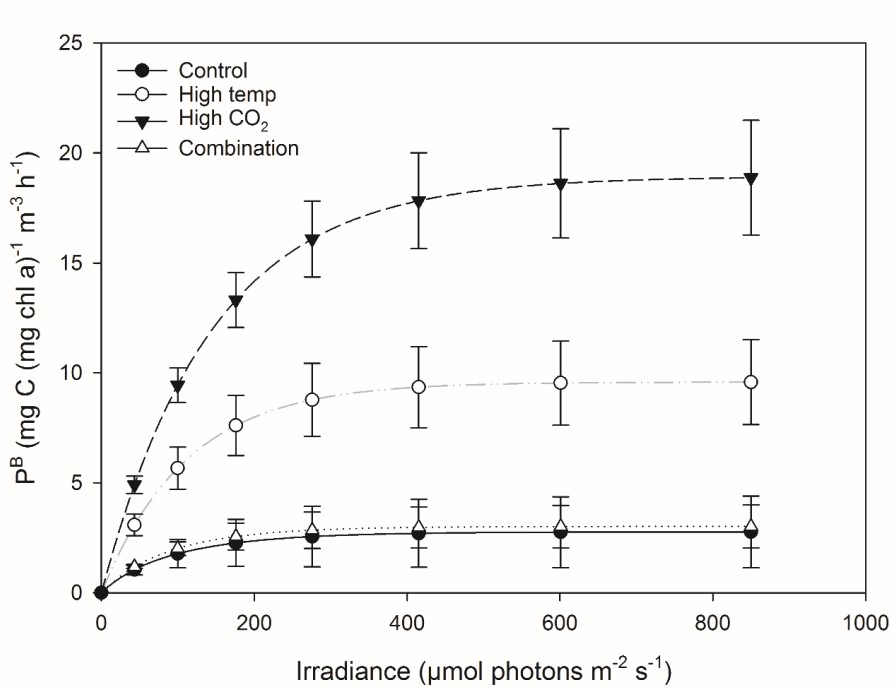

**Fig. 6.** Fitted parameters of FRRf-based photosynthesis-irradiance curves for the experimental treatments on the final experimental day (T36)

.036

.037

.038

.039



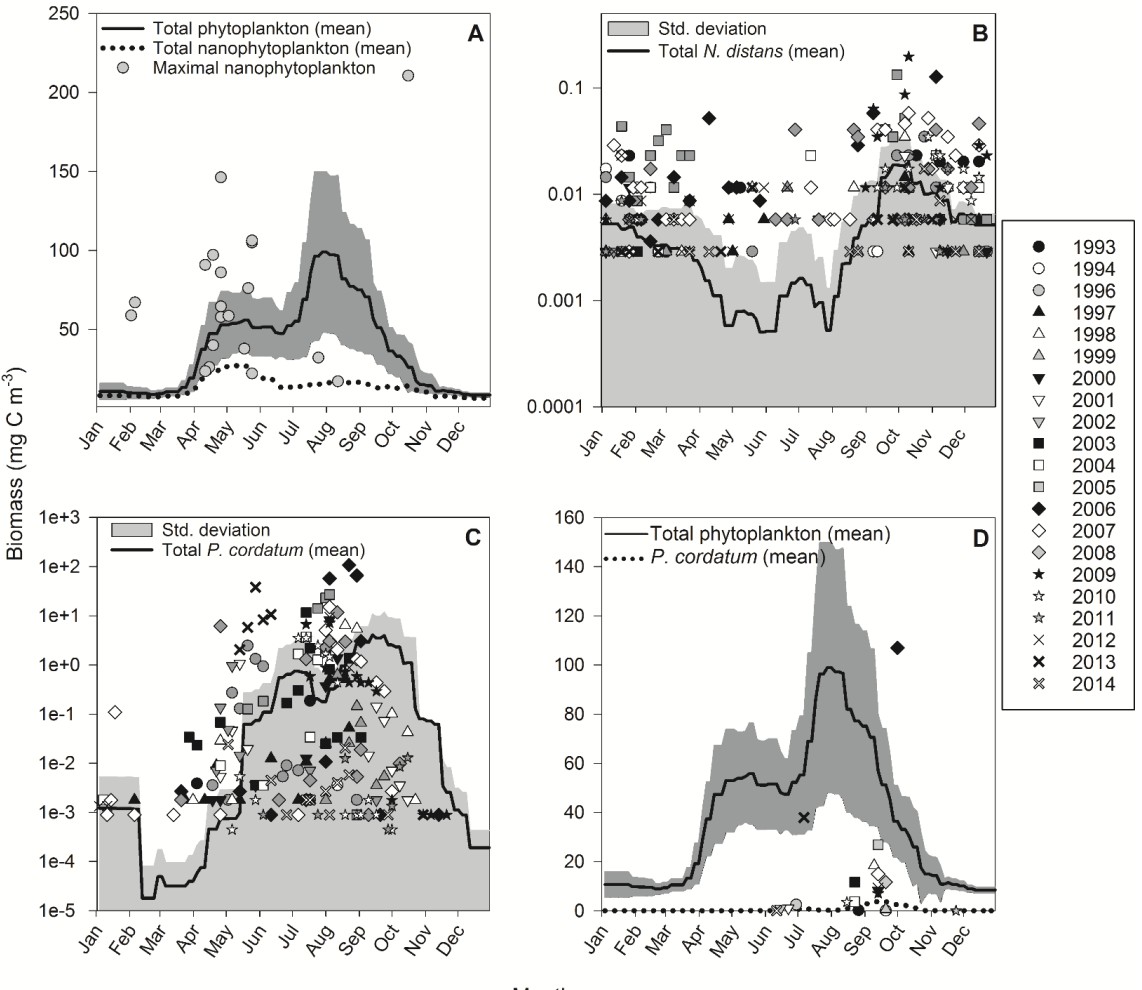

**Fig. 7.** (**A**) Temporal weekly profile of total phytoplankton carbon biomass at station L4 between 1993-2014. Black line is smoothed running average of total phytoplankton, grey area is standard deviation, dotted line is smoothed running average of total nanophytoplankton biomass and grey circles are maximal nanophytoplankton biomass from weekly observations (mean maxima of 70 mg C m$^{-3}$). (**B**) Seasonal profiles of *Navicula distans* (common log scale) between 1993-2014. Black line is smoothed running average over the time series, grey area is standard deviation and all symbols are observed data values by year. (**C**) Seasonal profiles of *Prorocentrum cordatum* (common log scale) between 1993-2014. Black line is smoothed running average over the time series, grey area is standard deviation and all symbols are observed data values by year. (**D**)Maximal *P. cordatum* biomass values relative to total phytoplankton biomass. Black line is smoothed running average of total phytoplankton biomass, grey area is standard deviation, dotted line is mean *P. cordatum* biomass and symbols are maximal *P. cordatum* biomass from weekly observations by year (as per figure legend for **B & C.**).

.040

.041

.042

.043





**Fig. 8.** Frequency distribution of biomass at station L4 along the *in-situ* gradients of temperature (1993-2014) and pCO₂ (2008-2014) for nanophytoplankton (**A & D**) *N. distans* (**B & E**) and *P. cordatum* (**C & F**).



**Table 1.** Results of generalised least-squares model testing for main effects of time, high temperature, high $CO_2$
and all interactions on chl *a*, phytoplankton biomass and particulate organic nitrogen. Significant results are in
bold; * $p < 0.05$, ** $p < 0.001$, *** $p < 0.0001$.

| Response variable | n | *df* | *t*-value | *p* | sig |
|---|---|---|---|---|---|
| **Chla (mg m⁻³)** | | | | | |
| Time | 80 | 72 | 3.782211 | **0.0003** | ** |
| High temp | 80 | 72 | 0.688339 | 0.4935 | |
| High $CO_2$ | 80 | 72 | 0.765811 | 0.4463 | |
| Time x high temp | 80 | 72 | 0.330431 | 0.742 | |
| Time x high $CO_2$ | 80 | 72 | -0.596962 | 0.5524 | |
| High temp x high $CO_2$ | 80 | 72 | 0.338096 | 0.7363 | |
| Time x high temp x high $CO_2$ | 80 | 72 | 1.302498 | 0.1969 | |
| **Estimated biomass (mg C m⁻³)** | | | | | |
| Time | 80 | 72 | 3.339498 | **0.0013** | * |
| High temp | 80 | 72 | -0.144359 | 0.8856 | |
| High $CO_2$ | 80 | 72 | -1.008942 | 0.3164 | |
| Time x high temp | 80 | 72 | 3.189888 | **0.0021** | * |
| Time x high $CO_2$ | 80 | 72 | 4.751901 | **0.0000** | *** |
| High temp x high $CO_2$ | 80 | 72 | 0.341905 | 0.7334 | |
| Time x high temp x high $CO_2$ | 80 | 72 | 0.449075 | 0.6547 | |
| **POC (mg m⁻³)** | | | | | |
| Time | 48 | 40 | -0.27037 | 0.7883 | |
| High temp | 48 | 40 | -1.2607 | 0.2147 | |
| High $CO_2$ | 48 | 40 | -1.13796 | 0.2619 | |
| Time x high temp | 48 | 40 | 5.31006 | **0.0000** | *** |
| Time x high $CO_2$ | 48 | 40 | 6.24182 | **0.0000** | *** |
| High temp x high $CO_2$ | 48 | 40 | -0.38194 | 0.7045 | |
| Time x high temp x high $CO_2$ | 48 | 40 | 1.21692 | 0.2308 | |
| **PON (mg m⁻³)** | | | | | |
| Time | 48 | 40 | 0.276438 | 0.7836 | |
| High temp | 48 | 40 | -1.447791 | 0.1555 | |
| High $CO_2$ | 48 | 40 | -1.571726 | 0.1239 | |
| Time x high temp | 48 | 40 | 4.78625 | **0.0000** | *** |
| Time x high $CO_2$ | 48 | 40 | 5.493647 | **0.0000** | *** |
| High temp x high $CO_2$ | 48 | 40 | 0.95334 | 0.3461 | |
| Time x high temp x high $CO_2$ | 48 | 40 | -0.126291 | 0.9001 | |
| **POC:PON (mg m⁻³)** | | | | | |
| Time | 48 | 40 | -3.248155 | **0.0024** | * |
| High temp | 48 | 40 | -0.206777 | 0.8372 | |
| High $CO_2$ | 48 | 40 | -0.055976 | 0.9556 | |
| Time x high temp | 48 | 40 | 2.433457 | **0.0195** | * |
| Time x high $CO_2$ | 48 | 40 | 3.838128 | **0.0004** | *** |
| High temp x high $CO_2$ | 48 | 40 | -2.932253 | **0.0055** | * |
| Time x high temp x high $CO_2$ | 48 | 40 | 2.40294 | **0.021** | * |




**Table 2.** Results of generalised linear model testing for significant effects of temperature, $CO_2$ and temperature x $CO_2$ on chl $a$ and phytoplankton biomass at the experiment end (T36). Significant results are in bold; * $p < 0.05$, ** $p < 0.001$, *** $p < 0.0001$.

| Response variable | n | df | z-value | p | sig |
|---|---|---|---|---|---|
| **Chl $a$ mg m$^{-3}$** | | | | | |
| High temp | 16 | 12 | 7.413 | **< 0.0001** | *** |
| High $CO_2$ | 16 | 12 | 0.804 | 0.437 | |
| High temp x high $CO_2$ | 16 | 12 | 18.043 | **<0.0001** | *** |
| **Total biomass (mg C m$^{-3}$)** | | | | | |
| High temp | 16 | 12 | 28.953 | **< 0.0001** | *** |
| High $CO_2$ | 16 | 12 | 36.042 | **< 0.0001** | *** |
| High temp x high $CO_2$ | 16 | 12 | 5.899 | **< 0.0001** | *** |
| **Diatoms (mg C m$^{-3}$)** | | | | | |
| High temp | 16 | 12 | -4.43 | **<0.0001** | *** |
| High $CO_2$ | 16 | 12 | 3.036 | **0.0024** | ** |
| High temp x high $CO_2$ | 16 | 12 | -7.243 | **<0.0001** | *** |
| **Dinoflagellates (mg C m$^{-3}$)** | | | | | |
| High temp | 16 | 12 | 9.848 | **<0.0001** | *** |
| High $CO_2$ | 16 | 12 | 1.805 | 0.2927 | |
| High temp x high $CO_2$ | 16 | 12 | 11.902 | **<0.0001** | *** |
| **Nanophytoplankton (mg m$^{-3}$)** | | | | | |
| High temp | 16 | 12 | 32.9 | **<0.0001** | *** |
| High $CO_2$ | 16 | 12 | 39.04 | **<0.0001** | *** |
| High temp x high $CO_2$ | 16 | 12 | 5.22 | **<0.0001** | *** |
| **_Synechococcus_ (mg m$^{-3}$)** | | | | | |
| High temp | 16 | 12 | 7.045 | **<0.0001** | *** |
| High $CO_2$ | 16 | 12 | -0.091 | 0.928 | |
| High temp x high $CO_2$ | 16 | 12 | 10.739 | **<0.0001** | *** |
| **Picophytoplankton (mg m$^{-3}$)** | | | | | |
| High temp | 16 | 12 | 0.413 | **0.679486** | |
| High $CO_2$ | 16 | 12 | 2.02 | **0.043435** | * |
| High temp x high $CO_2$ | 16 | 12 | 3.773 | **<0.0001** | *** |
| **Coccolithophores (mg C m$^{-3}$)** | | | | | |
| High temp | 16 | 12 | 0.276 | 0.782 | |
| High $CO_2$ | 16 | 12 | -0.368 | 0.713 | |
| High temp x high $CO_2$ | 16 | 12 | -1.265 | 0.206 | |
| **Cryptophytes (mg C m$^{-3}$)** | | | | | |
| High temp | 16 | 12 | 0.404 | 0.686 | |
| High $CO_2$ | 16 | 12 | 0.273 | 0.785 | |
| High temp x high $CO_2$ | 16 | 12 | 1.341 | 0.18 | |




**Table 3.** FRRf-based photosynthesis-irradiance curve parameters for the experimental treatments on the final
day (T36).

| Parameter | Control | sd | High temp | sd | High CO$_2$ | sd | Combination | sd |
|---|---|---|---|---|---|---|---|---|
| P$^B_m$ | 2.77 | 1.63 | 9.58 | 1.94 | 18.93 | 2.65 | 3.02 | 0.97 |
| α | 0.03 | 0.01 | 0.09 | 0.01 | 0.13 | 0.01 | 0.04 | 0.00 |
| $I_k$ | 85.33 | 45.47 | 110.93 | 6.09 | 144.13 | 17.91 | 86.38 | 33.06 |



**Table 4.** Results of generalised linear model testing for significant effects of temperature, CO$_2$ and temperature
x CO$_2$ on phytoplankton photophysiology; P$^B_m$ (maximum photosynthetic rates), α (light limited slope) and $I_k$
(light saturated photosynthesis). Significant results are in bold; * $p < 0.05$, ** $p < 0.001$, *** $p < 0.0001$.

| Response variable | n | df | t-value | p | sig |
|---|---|---|---|---|---|
| **P$^B_m$** | | | | | |
| High temp | 12 | 8 | 7.353 | **< 0.0001** | *** |
| High pCO$_2$ | 12 | 8 | 8.735 | **< 0.0001** | *** |
| High temp x high pCO$_2$ | 12 | 8 | -8.519 | **< 0.0001** | *** |
| | | | | | |
| **α** | | | | | |
| High temp | 12 | 8 | 13.03 | **< 0.0001** | *** |
| High pCO$_2$ | 12 | 8 | 15.15 | **< 0.0001** | *** |
| High temp x high pCO$_2$ | 12 | 8 | -14.82 | **< 0.0001** | *** |
| | | | | | |
| **$I_k$** | | | | | |
| High temp | 12 | 8 | 2.018 | 0.0783 | |
| High pCO$_2$ | 12 | 8 | 2.541 | **0.0347** | * |
| High temp x high pCO$_2$ | 12 | 8 | -2.441 | **0.0405** | * |


