# Peer review of "Effects of elevated CO$_2$ and temperature on phytoplankton community biomass, species composition and photosynthesis during an autumn bloom in the Western English Channel"

_Biogeosciences, 2017_

## Referee Comment (RC1) · Anonymous Referee #1 · 20 Dec 2017

The manuscript from Keys and collaborators deals with the impacts of ocean acidification and warming on the composition and biomass of the phytoplankton community in Autumn in the Western English Channel. The authors conducted a 36-day experiment in microcosms filled with seawater sampled in the declining phase of the autumn phytoplankton bloom at station L4 on October 7th 2015, where a long-term dataset of nutrient, chlorophyll a and community composition (among many other parameters) is available. This sampled seawater (sieved onto 200 microm) was used to fill 16 borosilicate bottles (2.5 L) corresponding to 4 replicates of 4 treatments. Treatments were

1) control (no modification of pH, T of 14.5 °C), 2) high CO2 (800 microatm, T of 14.5°C), 3) high T (ambient pCO2, T of 18.6 °C), and 4) high CO2 and T. The high T treatment appeared to be applied at once (seawater placed in a temperature regulated outdoor incubation system, while pCO2 was increased gradually to 800 microatm over 8 days. Each bottle was linked to reservoirs filled with filtered seawater in which nutrient concentrations were modified (NOx from 0.24 microM in situ to 8 microM, PO4 from 0.09 microM in situ to 0.5 microM, silicate maintained at in situ concentrations). pCO2 were also controlled in the reservoir for high pCO2 bottles in order to maintain constant pCO2 in the bottles. After 2 days conducted in batch mode, 10-13% of each experimental bottle were replaced with the medium contained in the reservoirs. Various parameters were sampled during the experiment at different frequencies. While chlorophyll a and carbonate parameters were sampled every 2-3 days, phytoplankton community biomass (biomass calculated based on flow cytometry data) was estimated on 5 occasions (T0, T10, T17, T24 and T36), and POC/PON were measured on 3 occasions (T0, T15, T36). Finally, the photosynthetic efficiency was investigated based on samples taken only at the end of the experiment (T36). Based on this experiment, the authors conclude that 1) in all treatments the phytoplankton community shifted from dinoflagellates to nanophytoplankton, 2) large nano-flagellates dominated in the control treatment, while smaller species dominated in the high CO2 and high T treatments, 3) combining these 2 "stressors", led to a different community with a higher proportion of dinoflagellates and especially of the HAB species Prorocentrum cordatum and 4) finally the authors conclude that "future increases in temperature and pCO2 do not appear to influence coastal phytoplankton productivity during autumn in the WEC which would have a negative feedback on atmospheric CO2".

Although this manuscript deals with the very important question being "how coastal phytoplankton will respond to global anthropogenic stressors and will coastal plankton community exert a feedback on atmospheric CO2 increase and global warming", I definitely cannot recommend this manuscript for Biogeosciences in its present form. My main concerns are 1) related to the experimental protocol considered and how it

could be used as a tool for projecting future evolutions of coastal plankton community in this area, 2) the way data have been used and conclusions that have been drawn based on those data, and 3) the way the manuscript is organized mixing results from an experiment and long-term in situ data, with both parts being in my opinion poorly related.

1) Experimental set-up

The authors mention that the effects of pCO2 and temperature on phytoplankton succession in autumn is presently unknown, which is the reason for their experimental investigation. However, I am really concerned about the experimental choices that have been made and I would like the authors to explain much better the rationale behind these choices. First of all, seawater was sampled at the end (let's say the declining phase) of the autumn bloom leading to low nitrate concentrations of 0.2 microM. My question is, is it realistic to force this system again to high levels of nitrate (8 microM, while the long-term average of NOx at station L4 is of 4.1 microM in October/November)? Don't you think your results are biased because of this, and prevent you from extrapolating your experimental results to the "real world"? The second experimental choice that is of concern to me is that, during the whole experiment, while chlorophyll a concentrations vary from 0.1 to 3-7 microg/L, the conditions of the carbonate chemistry have been maintained constant. Again, I do not understand the reason behind this choice. In situ, surface pCO2 is extremely dependent on biological activity (by far the main reason for the ocean being a sink of atmospheric CO2). So I have the same question that really needs to be fairly discussed in the paper: Don't you think your results are biased because of this, and prevent you to extrapolate your experimental results to the "real world"? Related to this, it seems to me that nutrient concentrations have been maintained constant during the experiment, although absolutely no data of nutrient concentrations are shown in the manuscript, which is not acceptable to me. Certainly much more than C availability, N and P (and Si) availability structure the composition of phytoplankton communities and control their productivity. What is

the reason for maintaining these parameters constant? Another experimental choice is to sieve the sampled seawater onto 200 microm, removing mesozooplankton grazers. Grazing is certainly a very important process shaping phytoplankton communities, what are the consequences of this choice, does it hamper your conclusions? Another missing (in my opinion) very important compartment is the heterotrophic prokaryotic community for which no data are shown in the manuscript (bacterial abundance at least could help). Finally, concerning these experimental choices, why did you conduct an experiment over 36 days? I will come back on that later in the second part, but this choice needs to be explained. This is a very important point since you base a lot of your conclusions on the interpretation of results obtained at T36 only.

2) Data analysis

The authors considered an experimental set-up in which they sampled their experimental bottles on a regular (yet variable depending on the parameter) basis. Nonetheless, many of their conclusions are based only on the analysis of data obtained at T36. For instance, they discuss on L456 to 462, that "chlorophyll a was significantly higher in the combination treatment at T36, . . ., but Chl a was significantly lower in the high pCO2 treatment. . .". I apologize but this interpretation does not make sense at all. Actually, T36 seems to be the only sampling point for which these conclusions are valid. At the penultimate sampling point, there is as much Chl a in the combination treatment than in the high CO2 while high T and control lead to lower concentrations. If you choose another time point, you reach another conclusion. . . and so on. . . The entire dataset MUST be used instead of single points. This is actually even more problematic for parameters that have been sampled with a lower frequency (especially POC and PON), again conclusions have been drawn based on the last sampling point. At T10, the biomass is the highest in the high T, followed by high T/high COP2, then high CO2 and control with the lowest biomass. At T17, you can draw another conclusion etc. . . This is also true for community composition, for which you insist a lot on the abrupt increase in dinoflagellate abundance in the high T/high CO2 treatment at T36, while

their abundance is much lower at T24.... Again, what is the rationale in using data obtained after 36 days of incubation in a small volume with all the artefacts associated with this incubation technique...?

3) Structure of the manuscript

I am not convinced about the way this manuscript that combines analyses of in situ long-term data and experimental data, is structured. In my opinion, the analysis of in situ data should be put upfront in the manuscript, before describing and analyzing the experimental results. Furthermore, I am not convinced about the relevance of these observational data in this manuscript. Analyzing the distribution of the abundance of phytoplankton species just based on temperature and pCO2 is a huge simplification.

Minor comments.

Introduction. L36 to 58. Citing 10 years old (at least) papers in this section is not acceptable and suggests (wrongly I suppose) that the authors are not aware of recent literature. L43: please add "surface" pH of 0.3 units... L81: what is the reason of this warming? L83: this sentence makes no sense. If no significant trend, there is no increase....

Mat and Met L131: what were the PAR levels during the experiment?

Results L254: please add: "with a mean concentration over this period, of ... or equivalent Please provide data of nutrients and data of TA and DIC. Regarding TA and DIC, I am extremely surprised by Figure 2. How can you have similar pCO2/pH and $CO_3^{2-}$/$HCO_3^{2-}$ between control T and high T? With a difference of 4 °C, this appears unreal.... If you keep pCO2 constant as you mention, you should have several hundreds of microM difference between $HCO_3^-$ at 14°C compared to 18°C.... or am I wrong?

On several occasions, please add "atmospheric CO2 increase" when you refer to the potential negative or positive feedbacks on atmospheric CO2.

---

## Referee Comment (RC2) · Anonymous Referee #2 · 12 Jan 2018

The manuscript 'Effects of elevated CO$_2$ and temperature on phytoplankton community biomass, species composition and photosynthesis during an autumn bloom in the Western English Channel' by Keys et al. presents much needed data on the combined effects of two stressors of ocean change on the base of the marine food-web. Furthermore, a twenty year long monitoring record of phytoplankton community composition at the experimental site is shown. However, in its present form, I do not agree with the way the experimental data was analyzed and hence with certain conclusions.

Major comments and suggestions:

[Figure]

1) The time-series data on the natural variability of phytoplankton community composition and biomass does not complement the experimental data set and is actually disconnected. It is not clear how both data sets would complement each other which is also reflected in the fact that the time-series data is not mentioned in the abstract. It hence appears to be an unnecessary add-on.

2) Most critically, however, I consider the choice of the authors to restrict their analysis of the experiments to the last data point at the end of incubations on day 36 as being problematic. This appears to be arbitrary as many of the main conclusions would be different for the sampling days before, and most likely for the ones to come, if the experiments would have been run for a longer period of time. A different approach is needed.

3) I found it interesting that phytoplankton biomass in the combined high $CO_2$ and temperature treatment, and the control did decrease from day 20 and 25 onwards (Fig. 3B). Why is this not reflected in Chl a development and what is the cause? In the semi-continuous culturing set-up of the experiments with a daily dilution of about 10% of the incubation volume with fresh media containing 8 and 0.5 $L^{-1}$ nitrate and phosphate, respectively, a decline in phytoplankton biomass suggests that net growth has slowed down and is lower than the dilution rate. The PON data, however, suggests that there should be ample amounts of dissolved inorganic nutrients left for phytoplankton growth, thus it appears that there are indirect effects at work which should be discussed.

4) The photosynthesis versus irradiation curves for the four treatments are based on the assumption that the electron requirement for carbon uptake is independent of seawater $CO_2$ concentration, temperature and species composition. Since this is most likely not the case, any conclusions drawn (if any) would need to be discussed with much more caution.

Additional comments and suggestions:

1) P2, L36: I assume you mean 'concentrations of $CO_2$', not 'uptake of atmospheric

CO$_2$'.

2) P2 L63-66: The grammar in the last sentence of this page seems wrong.

3) P3, L70: Citing only Engel et al. (2008) and Moutaka-Gouni et al. (2016) here is very selective.

4) P4, L127: It should read 'was' not 'were'.

5) P6, L170: Strictly speaking, the calculation of carbonate system parameters from DIC and TA require to account for the contribution of silicate and phosphate to the latter. Have dissolved inorganic nutrients been measured in the incubations?

6) P6, L187: How was size determined for the particles measured by flow cytometry, e.g. forward scatter calibrated with fluorescent beads of known size?

7) P14, L446: Most phytoplankton also uses CO$_2$ as an inorganic carbon source, not only HCO$_3^-$. Furthermore, 'more efficient' use of CO$_2$ in comparison to what? Finally, I did not understand the rational behind the notion that *Phaeocystis* would have an advantage at higher CO$_2$ levels.

8) P14, L458: I agree, but what could be an explanation for this finding (see also comment above)?

9) P14, L467: The four references on CO$_2$ effects on phytoplankton community biomass appear to be a very selective choice. Furthermore, this paragraph lacks any conclusions.

10) P16, L501: The temperature for maximum photosynthetic rates should be species specific.

11) P16, L517: I would assume that almost any autotrophic organisms growing at pH levels above 9 would slow down in growth because of inorganic carbon limitation, not only dinoflagellates.

12) P18, L578-609: This discussion on *Prorocentrum* is unconnected to the experimental data and I do not see any added benefit or conclusions to be drawn.

13) P19, L618: It is not clear to me what the authors mean with the 'present upper limit of the $pCO_2$ threshold increase'.

14) P20, L648: Why are there potential positive feedbacks?

15) P20, L651: Why is there a potential for negative impacts on ecosystem functioning?

16) P20, L556: Why do 'little response and no effects' suggest 'negative feedbacks'?

---

## Author Comment (AC1) · 2 Feb 2018

Dear reviewer and editors,

The authors would like to thank the reviewer for the helpful comments provided in order to improve our manuscript. The manuscript has been revised based on the reviewers feedback. Our responses to the detailed comments are listed below.

The manuscript from Keys and collaborators deals with the impacts of ocean acidification and warming on the composition and biomass of the phytoplankton community

[Figure]

in Autumn in the Western English Channel. The authors conducted a 36-day experiment in microcosms filled with seawater sampled in the declining phase of the autumn phytoplankton bloom at station L4 on October 7th 2015, where a long-term dataset of nutrient, chlorophyll a and community composition (among many other parameters) is available. This sampled seawater (sieved onto 200 microm) was used to fill 16 borosilicate bottles (2.5 L) corresponding to 4 replicates of 4 treatments. Treatments were 1) control (no modification of pH, T of 14.5 _C), 2) high $CO_2$ (800 microatm, T of 14.5_C), 3) high T (ambient $pCO_2$, T of 18.6 _C), and 4) high $CO_2$ and T. The high T treatment appeared to be applied at once (seawater placed in a temperature regulated outdoor incubation system, while $pCO_2$ was increased gradually to 800 microatm over 8 days. Each bottle was linked to reservoirs filled with filtered seawater in which nutrient concentrations were modified (NOx from 0.24 microM in situ to 8 microM, PO4 from 0.09 microM in situ to 0.5 microM, silicate maintained at in situ concentrations). $pCO_2$ were also controlled in the reservoir for high $pCO_2$ bottles in order to maintain constant $pCO_2$ in the bottles. After 2 days conducted in batch mode, 10-13% of each experimental bottle were replaced with the medium contained in the reservoirs. Various parameters were sampled during the experiment at different frequencies. While chlorophyll a and carbonate parameters were sampled every 2-3 days, phytoplankton community biomass (biomass calculated based on flow cytometry data) was estimated on 5 occasions (T0, T10, T17, T24 and T36), and POC/PON were measured on 3 occasions (T0, T15, T36). Finally, the photosynthetic efficiency was investigated based on samples taken only at the end of the experiment (T36). Based on this experiment, the authors conclude that 1) in all treatments the phytoplankton community shifted from dinoflagellates to nanophytoplankton, 2) large nano-flagellates dominated in the control treatment, while smaller species dominated in the high $CO_2$ and high T treatments, 3) combining these 2 "stressors", led to a different community with a higher proportion of dinoflagellates and especially of the HAB species Prorocentrum cordatum and 4) finally the authors conclude that "future increases in temperature and $pCO_2$ do not appear to influence coastal phytoplankton productivity during autumn in the WEC

which would have a negative feedback on atmospheric CO2". Although this manuscript deals with the very important question being "how coastal phytoplankton will respond to global anthropogenic stressors and will coastal plankton community exert a feedback on atmospheric CO2 increase and global warming", I definitely cannot recommend this manuscript for Biogeosciences in its present form. My main concerns are 1) related to the experimental protocol considered and how it could be used as a tool for projecting future evolutions of coastal plankton community in this area, 2) the way data have been used and conclusions that have been drawn based on those data, and 3) the way the manuscript is organized mixing results from an experiment and long-term in situ data, with both parts being in my opinion poorly related. 1) Experimental set-up The authors mention that the effects of pCO2 and temperature on phytoplankton succession in autumn is presently unknown, which is the reason for their experimental investigation. However, I am really concerned about the experimental choices that have been made and I would like the authors to explain much better the rationale behind these choices. First of all, seawater was sampled at the end (let's say the declining phase) of the autumn bloom leading to low nitrate concentrations of 0.2 microM. My question is, is it realistic to force this system again to high levels of nitrate (8 microM, while the long-term average of NOx at station L4 is of 4.1 microM in October/ November)? Don't you think your results are biased because of this, and prevent you from extrapolating your experimental results to the "real world"? The second experimental choice that is of concern to me is that, during the whole experiment, while chlorophyll a concentrations vary from 0.1 to 3-7 microg/L, the conditions of the carbonate chemistry have been maintained constant. Again, I do not understand the reason behind this choice. In situ, surface pCO2 is extremely dependent on biological activity (by far the main reason for the ocean being a sink of atmospheric CO2). So I have the same question that really needs to be fairly discussed in the paper: Don't you think your results are biased because of this, and prevent you to extrapolate your experimental results to the "real world"? Related to this, it seems to me that nutrient concentrations have been maintained constant during the experiment, although absolutely no data of nutrient

concentrations are shown in the manuscript, which is not acceptable to me. Certainly much more than C availability, N and P (and Si) availability structure the composition of phytoplankton communities and control their productivity. What is the reason for maintaining these parameters constant? Another experimental choice is to sieve the sampled seawater onto 200 microm, removing mesozooplankton grazers. Grazing is certainly a very important process shaping phytoplankton communities, what are the consequences of this choice, does it hamper your conclusions? Another missing (in my opinion) very important compartment is the heterotrophic prokaryotic community for which no data are shown in the manuscript (bacterial abundance at least could help). Finally, concerning these experimental choices, why did you conduct an experiment over 36 days? I will come back on that later in the second part, but this choice needs to be explained. This is a very important point since you base a lot of your conclusions on the interpretation of results obtained at T36 only.

Response - It is realistic to force this system from low levels (0.2 microM) to high levels (8 microM) of nitrate, which can frequently occur after heavy rainfall during autumn (see Barnes et al. 2015a; Fig 6) from August to December. In addition, during a pilot study experiment in which we kept nitrate low in all treatments, the phytoplankton populations in all treatments crashed within 7 days. Of course, it depends on what the research question is and for these experiments we are asking: what the long-term trend is to elevated CO2 and temperature. We are not asking the question about adaptation to nutrients, hence why it has been kept replete in all treatments. This has now been highlighted in the methods section. This research was from a PhD thesis. Whilst I had the expertise and resources to measure the carbonate chemistry, biological and photo-physiological parameters during the experiments I did not have the resources or training to measure nutrients for each treatment, replicate and time point over the course of the experiment, hence why no data of nutrient concentrations are shown in the manuscript (following T0). This has now been highlighted in the methods section.

The seasonality in pH and TA are fairly stable at L4 with high pH and low DIC during

summer, and low pH, high DIC during winter (Kitidis et al. 2012 CSR), whereas Chl a is much more variable with high values in spring and summer and low values in both summer and winter (see Smyth et al. 2010; Widdicombe et al. 2010; Barnes et al. 2015a). By maintaining the carbonate chemistry, we are mimicking natural events at L4. This has now been highlighted in the discussion section.

We agree with the reviewer that grazing could certainly be important in shaping the phytoplankton community, however our question was, what were the combined effects of CO2 and temperature on phytoplankton biomass and photosynthesis, not what were the combined effects of CO2 and temperature on zooplankton grazing. This has now been highlighted in the methods section.

We did not sample for bacterial abundance.

To provide sufficient time for changes in the phytoplankton community to occur and to achieve an ecologically relevant data set, a primary experimental goal was to extend the incubation period beyond the short-term acclimation phase. Exactly how long the acclimation phase is with natural populations may be the subject of some debate. However, previous pilot experiments using the same experimental protocols (unpublished data) have highlighted that after 24 days of incubation, significant changes in community structure and increases in biomass were observed. These pilot study results were used to inform a more relevant incubation period (i.e. 36 days).

2) Data analysis The authors considered an experimental set-up in which they sampled their experimental bottles on a regular (yet variable depending on the parameter) basis. Nonetheless, many of their conclusions are based only on the analysis of data obtained at T36. For instance, they discuss on L456 to 462, that "chlorophyll a was significantly higher in the combination treatment at T36, : : :, but Chl a was significantly lower in the high pCO2 treatment: : :". I apologize but this interpretation does not make sense at all. Actually, T36 seems to be the only sampling point for which these conclusions are valid. At the penultimate sampling point, there is as much Chl a in the combination treatment

than in the high CO2 while high T and control lead to lower concentrations. If you choose another time point, you reach another conclusion: : : and so on: : : The entire dataset MUST be used instead of single points. This is actually even more problematic for parameters that have been sampled with a lower frequency (especially POC and PON), again conclusions have been drawn based on the last sampling point. At T10, the biomass is the highest in the high T, followed by high T/high COP2, then high CO2 and control with the lowest biomass. At T17, you can draw another conclusion etc: : : This is also true for community composition, for which you insist a lot on the abrupt increase in dinoflagellate abundance in the high T/high CO2 treatment at T36, while their abundance is much lower at T24: : :. Again, what is the rationale in using data obtained after 36 days of incubation in a small volume with all the artefacts associated with this incubation technique: : :?

Response - We presented analysis in the form of generalised least squares model results on all of the data (i.e. at all time points) for chlorophyll a concentrations (L 284), estimated total biomass (L 296 – L298), POC, PON and POC:PON (L307 – L317). The results of these analyses were also presented in Table 1 (page 40). However, we wanted to illustrate the abrupt regime shifts in community composition between T24 and T36, when the control community switched to diatoms, and the combination treatment (elevated pCO2/elevated temperature) switched to the most diverse experimental community with significant contributions from dinoflagellates and Synechococcus. For this reason, we have focussed the community structure analysis on differences in composition between experimental treatments at T36. Critically, this highlights the importance of the experimental incubation period (i.e. extending to 36 days following significant shifts in taxonomic composition and biomass beyond 24 days in this study and in a previous pilot study (data not published). We have updated the manuscript to provide additional analysis on all time points to evaluate the relative shifts of the community through time across the different treatments.

3) Structure of the manuscript I am not convinced about the way this manuscript that

combines analyses of in situ long-term data and experimental data, is structured. In my opinion, the analysis of in situ data should be put upfront in the manuscript, before describing and analyzing the experimental results. Furthermore, I am not convinced about the relevance of these observational data in this manuscript. Analyzing the distribution of the abundance of phytoplankton species just based on temperature and pCO2 is a huge simplification.

Response - We agree that analysis of the time-series natural variability based on temperature and pCO2 alone is a simplification of a complex natural system. We have restructured the manuscript by shifting portions of this section describing the natural variability at the L4 study site, ahead of the microcosm experiment in the introduction. Any further analysis and reference to the time-series in the results/discussion section has now been removed from the manuscript.

Minor comments. Introduction. L36 to 58. Citing 10 years old (at least) papers in this section is not acceptable and suggests (wrongly I suppose) that the authors are not aware of recent literature. L43: please add "surface" pH of 0.3 units: : : L81: what is the reason of this warming? L83: this sentence makes no sense. If no significant trend, there is no increase: : :. Mat and Met L131: what were the PAR levels during the experiment? Results L254: please add: "with a mean concentration over this period, of : : : or equivalent Please provide data of nutrients and data of TA and DIC. Regarding TA and DIC, I am extremely surprised by Figure 2. How can you have similar pCO2/pH and CO32-/HCO32- between control T and high T? With a difference of 4 _C, this appears unreal: : :. If you keep pCO2 constant as you mention, you should have several hundreds of microM difference between HCO3- at 14_C compared to 18_C: : :. or am I wrong? On several occasions, please add "atmospheric CO2 increase" when you refer to the potential negative or positive feedbacks on atmospheric CO2.

Response - We have incorporated all of your minor comments into the manuscript. You have rightly pointed out that we presented the wrong CO32-/HCO32- data in error (Fig. 2). We thank you for drawing our attention to this error and have updated this figure
in the manuscript. We have also included an additional table of all carbonate system parameters in the supplementary material. We are however unable to provide further nutrient data as discussed above.

---

## Author Comment (AC2) · 2 Feb 2018

Dear reviewer and editors, The authors would like to thank the reviewer for the helpful comments provided in order to improve our manuscript. The manuscript has been revised based on the reviewers feedback. Our responses to the detailed comments are listed below.

The manuscript 'Effects of elevated CO2 and temperature on phytoplankton community biomass, species composition and photosynthesis during an autumn bloom in the

[Figure]

Western English Channel' by Keys et al. presents much needed data on the combined effects of two stressors of ocean change on the base of the marine food-web. Furthermore, a twenty year long monitoring record of phytoplankton community composition at the experimental site is shown. However, in its present form, I do not agree with the way the experimental data was analyzed and hence with certain conclusions. Major comments and suggestions: 1) The time-series data on the natural variability of phytoplankton community composition and biomass does not complement the experimental data set and is actually disconnected. It is not clear how both data sets would complement each other which is also reflected in the fact that the time-series data is not mentioned in the abstract. It hence appears to be an unnecessary add-on.

Response - The time-series in the results/discussion section has now been removed from the manuscript. We have restructured the manuscript by shifting portions of this section describing the natural variability at the L4 study site, ahead of the microcosm experiment in the introduction.

2) Most critically, however, I consider the choice of the authors to restrict their analysis of the experiments to the last data point at the end of incubations on day 36 as being problematic. This appears to be arbitrary as many of the main conclusions would be different for the sampling days before, and most likely for the ones to come, if the experiments would have been run for a longer period of time. A different approach is needed.

Response - We presented analysis in the form of generalised least squares model results on all of the data (i.e. at all time points) for chlorophyll a concentrations (L 284), estimated total biomass (L 296 – L298), POC, PON and POC:PON (L307 – L317). The results of these analyses were also presented in Table 1 (page 40). However, we wanted to illustrate the abrupt regime shifts in community composition between T24 and T36, when the control community switched to diatoms, and the combination treatment (elevated pCO2/elevated temperature) switched to the most diverse experimental community with significant contributions from dinoflagellates and Synechococcus.

For this reason, we have focussed the community structure analysis on differences in composition between experimental treatments at T36. Critically, this highlights the importance of the experimental incubation period (i.e. extending to 36 days following significant shifts in taxonomic composition and biomass beyond 24 days in this study and in a previous pilot study (data not published). We have updated the manuscript to provide additional analysis on all time points to evaluate the relative shifts of the community through time across the different treatments.

3) I found it interesting that phytoplankton biomass in the combined high CO2 and temperature treatment, and the control did decrease from day 20 and 25 onwards (Fig. 3B). Why is this not reflected in Chl a development and what is the cause? In the semicontinuous culturing set-up of the experiments with a daily dilution of about 10% of the incubation volume with fresh media containing 8 and 0.5 LôĂĂĂ1 nitrate and phosphate, respectively, a decline in phytoplankton biomass suggests that net growth has slowed down and is lower than the dilution rate. The PON data, however, suggests that there should be ample amounts of dissolved inorganic nutrients left for phytoplankton growth, thus it appears that there are indirect effects at work which should be discussed.

Response – Biomass in the control peaked at T25 followed by decline to T36. In line with this biomass trend, Chl a also peaked at T25 in the control (3.9 mg m-3) and declined to 3.3 mg m-3 by T27, remaining close to this value until T36. Biomass in the combination treatment (high CO2 and high temperature) peaked at T20 followed by decline to T36 whereas Chl a in this treatment declined from T20 (3.8 mg m-3) to T25 (3.1 mg m-3) followed by an increase at T27 (to 5.4 mg m-3) before further decline in line with biomass. Chl a peaked in this treatment again at T36 (6.8 mg m-3). We attribute the increase in Chl a between T25 – T27 (coincident with an overall biomass decrease) to lower species specific carbon:chlorophyll ratios (C:Chl a) since dinoflagellates, Synechococcus and picophytoplankton biomass increased in this treatment from T25. Sampling throughout the experiment was conducted by peristaltic pumps linked

to sampling tubes positioned in the centre of each incubation bottle. At T36 however, incubation bottles were removed from the incubator and samples were poured after gentle mixing directly from the incubation bottles. We cannot exclude the possibility of non-phytoplankton particles (e.g. build up on incubation bottle walls) influencing the final high Chl a value at T36 in this treatment. The manuscript has been amended to show C:Chl a data and this (including published literature on C:Chl a values) together with potential bottle effects have been added to the discussion. We further attribute declining biomass under nutrient replete conditions in the combination treatment to changes in community structure in the context of differential species-specific growth rates in respect of increasing dinoflagellate biomass.

4) The photosynthesis versus irradiation curves for the four treatments are based on the assumption that the electron requirement for carbon uptake is independent of seawater $CO_2$ concentration, temperature and species composition. Since this is most likely not the case, any conclusions drawn (if any) would need to be discussed with much more caution.

Response - We applied the same electron requirement parameter for carbon uptake across all treatments and we acknowledge that in nature and between species, there exists significant differences in this parameter (e.g. variation of 1.15 to 54.2 mol e- (mol C)-1, Lawrenz et al, 2013; Hancke et al, 2015) which co-vary with temperature, nutrients, Chl a, irradiance and community structure. We have amended our manuscript to consider this variability relative to our method and results within the discussion.

Additional comments and suggestions: 1) P2, L36: I assume you mean 'concentrations of CO2', not 'uptake of atmospheric CO2'.

Response – We have amended this sentence.

2) P2 L63-66: The grammar in the last sentence of this page seems wrong.

Response – We have amended this sentence.

3) P3, L70: Citing only Engel et al. (2008) and Moutaka-Gouni et al. (2016) here is very selective. Response – We have amended the manuscript to discuss further trends in the response of nano- and picophytoplankton with appropriate references to expand this paragraph.

4) P4, L127: It should read 'was' not 'were'.

Response – We have amended this sentence.

5) P6, L170: Strictly speaking, the calculation of carbonate system parameters from DIC and TA require to account for the contribution of silicate and phosphate to the latter. Have dissolved inorganic nutrients been measured in the incubations?

Response - This research was from a PhD thesis. I did not have the resources or training to measure nutrients for each treatment, replicate and time point over the course of the experiment, hence no data of nutrient concentrations are shown in the manuscript beyond T0. Since phosphate was amended and held constant at 0.5 $\mu$M L-1, this value was used in the calculation of carbonate system parameters.

6) P6, L187: How was size determined for the particles measured by flow cytometry, e.g. forward scatter calibrated with fluorescent beads of known size?

Response - Phytoplankton data acquisition was triggered on both chlorophyll fluorescence and forward light scatter (FSC) using prior knowledge of the position of Synechococcus sp. to set the lower limit of analysis. Density plots of FSC vs. CHL fluorescence, phycoerythrin fluorescence vs. CHL fluorescence and side scatter (SSC) vs. CHL fluorescence were used to discriminate Synechococcus sp., picoeukaryote phytoplankton (approx. 0.5–3 $\mu$m), coccolithophores, cryptophytes, Phaeocystis sp. single cells and nanophytoplankton (eukaryotes >3 $\mu$m, excluding the coccolithophores, cryptophytes and Phaeocystis sp. single cells).

7) P14, L446: Most phytoplankton also uses CO2 as an inorganic carbon source, not only HCOôĂĂ3 . Furthermore, 'more efficient' use of CO2 in comparison to what?

Finally, I did not understand the rational behind the notion that Phaeocystis would have an advantage at higher CO2 levels.

Response – We have amended this paragraph to better reflect our experimental findings that under elevated CO2, Phaeocystis spp. dominated the community and this is likely due to variability in the carbon acquisition strategy between the species present, in favour of Phaeocystis spp.

8) P14, L458: I agree, but what could be an explanation for this finding (see also comment above)?

Response – Please refer to our response to your major comment No. 3) above.

9) P14, L467: The four references on CO2 effects on phytoplankton community biomass appear to be a very selective choice. Furthermore, this paragraph lacks any conclusions.

Response – We have amended the manuscript to discuss further cited studies on CO2 effects and conclude upon the trends.

10) P16, L501: The temperature for maximum photosynthetic rates should be species specific.

Response – We have amended the manuscript to acknowledge species-specific maximum photosynthetic rates.

11) P16, L517: I would assume that almost any autotrophic organisms growing at pH levels above 9 would slow down in growth because of inorganic carbon limitation, not only dinoflagellates. Response – We have amended the manuscript to reflect effects of high pH on the growth of all phytoplankton species, not only dinoflagellates which were used in this particular section of the discussion. 12) P18, L578-609: This discussion on Prorocentrum is unconnected to the experimental data and I do not see any added benefit or conclusions to be drawn. Response – We have removed the discussion on Prorocentrum cordatum. 13) P19, L618: It is not clear to me what the authors

mean with the 'present upper limit of the pCO2 threshold increase'. Response – Since we have removed the time-series analysis from the manuscript, the section containing this line (p19, L611-645) has been removed. 14) P20, L648: Why are there potential positive feedbacks? Response – The increased photosynthetic rates observed in our individual treatments of elevated CO2 and elevated temperature suggest these single factors may lead to removal of more CO2 from the surface ocean than under current ambient conditions. This may therefore lead to a more rapid exchange of CO2 between the surface ocean-atmosphere boundary layer, leading to positive feedbacks on atmospheric CO2. We have updated the manuscript to make this statement clearer. 15) P20, L651: Why is there a potential for negative impacts on ecosystem functioning? Response - Dense blooms of Phaeocystis spp. in some ecosystems can be responsible for fish and shell-fish mortality (Levasseur et al, 1994; Peperzak & Poelman, 2008). Phaeocystis spp. colony mucous matrix can inhibit copepod grazing, and therefore affect food web structure through predator-prey size mis-match. Additionally, carbohydrates excreted by Phaeocystis spp. that coagulate to form transparent exopolymer particles (TEP) have strong inhibitory feeding effects on both nauplii and adult copepods (Dutz et al., 2005). Phaeocystis spp. can also be inadequate as a food source for some copepods (e.g. Calanus helgolandicus, Temora stylifera and Acartia tonsa), which can lead to negative effects on fecundity and egg production (Tang et al., 2001; Turner et al., 2002). Exotoxins produced by Phaeocystis spp. during the spring bloom in the northern Norwegian coast can also induce stress in cod larvae (Gadus morhua) (Eilertsen and Raa, 1995). Mass fish mortalities have been linked to Phaeocystis spp. blooms in the Irish Sea (Rogers and Lockwood, 2009) and south-eastern Vietnamese coastal waters (Tang et al., 2004). In addition, the odorous foam produced by Phaeocystis spp. blooms can wash up on beaches and create anoxic conditions in the surface sediment which can lead to mortality of the intertidal benthic community (Desroy and Denis, 2004; Spilmont et al., 2009). Our microcosm experiment suggests that future high CO2 scenarios could increase Phaeocystis spp. blooms at station L4 in the WEC which could adversely affect ecosystem functioning, food web structure and fisheries.

16) P20, L556: Why do 'little response and no effects' suggest 'negative feedbacks'?
Response – We assume you refer to L656 in relation to our statement on negative feedbacks. No significant increase in community biomass or photosynthetic rates were observed in our combination treatment of elevated CO2 and temperature. This suggests no change in the removal of CO2 via photosynthesis from the surface ocean relative to current ambient conditions. Under conditions of future increased atmospheric CO2, no change in the surface ocean uptake of CO2 would therefore lead to a negative feedback on atmospheric CO2 concentrations.

---

## Author Response (AR2)

**Authors response to reviewer's comments: Keys et al. BG-2017-510**

Dear authors,
Thank you for submitting a revised version of your manuscript. One of the original reviewers has examined it and considers that most comments and suggestions have been dealt with adequately. However, this reviewer lists numerous issues that still require attention before publication can be recommended. I also include below several points that need to be considered. Some of them are important, in particular those that concern 1) discussion of the implications of not having nutrient data, 2) interpretation of potential existence of feedback mechanisms, 3) comparison of properties between initial and final sampling date and 4) explanation of the antagonistic effect between high pCO2 and warm temperature.
I now invite you to address these comments in a revised version of your manuscript.
Thank you for submitting your work to Biogeosciences.

Best regards,
Emilio Marañón

Editorial comments on BG-2017-510

- As indicated by reviewer 1, the Discussion must consider the implications of the lack of information on nutrient concentration. The text should explicitly mention that differences in nutrient availability may have contributed to observed differences between control and high temperature and high CO2 treatments. The whole experiment rests on the assumption that initial nutrient concentrations were the same in every experimental bottle, but this assumption has not been verified. Assuming this was the case, and to the extent that differences in biomass were observed between treatments, differences in nutrient concentration must have arisen as well during the experiment.

*Response:* The text now explicitly mentions that differences in nutrient availability may have contributed to observed differences between control and high temperature and high CO2 treatments (see Lines 445-447).

- Title must be changed to make it clear that what is being studied is a bloom induced experimentally in vitro.

*Response:* The title has been changed as suggested.

- Abstract line First sentence should clarify that the article deals with an experimentally induced bloom in vitro.

*Response:* The first sentence in the abstract has been changed to indicate that the article deals with an experimentally induced bloom in vitro.

- In the Abstract, and throughout the manuscript, changes in properties as a result of experimental treatments must be described in terms of the difference between treatment and control, not between treatment and initial value. For instance, lines 16-17 read: 'total phytoplankton biomass was significantly increased by elevated pCO2 (20-fold) and (…) biomass also increased under elevated temperature (15-fold)'. This passage suggests that high pCO2 and warm temperature
induced very large increases in biomass, when in fact those increases (20- and 15-fold) refer to the
difference between final and initial values. Given that biomass increased also in the control bottles,
the relevant comparison is between treatments and control. The same problem applies to the later
reference to the 30-fold increase in Prorocentrum cordatum.

*Response: The manuscript has been modified to reflect differences between treatments rather than*
*differences within treatments over time.*

Line 19 Add 'and in the control': 'Throughout the experiment in all treatments and in the control…'
*Response:* The sentence has been modified as suggested.

Lines 107-109 Note that pre-screening through 200-um mesh removes only the mesozooplankton.
The microzooplankton (quantitatively more relevant, in terms of grazing pressure) are still present in
the experimental bottles (see comment 2 by reviewer 2).

*Response:* The sentence has been modified as suggested.

- The assessement of chla-specific productivity is based on a indirect method (FRRF), which, as the
authors acknowledge, is based on numerous untested assumptions. In addition, FRRF measurements
were conducted on a single occasion in a experiment that lasted >30 days. This represents a tenuous
base to make a major conclusion of the observation that the combined high pCO2 and high
temperature treatment causes no change in chla-normalised productivity.

*Response:* We have now clearly pointed out that the measurements were only made at the end of
the experiment and that robust conclusions cannot be made on this information alone.

- Related to the point above, and as mentioned also by reviewer 1, the lack of effect of the
combinated high pCO2 and high temperature treatment upon productivity is by no means an
instance of negative feedback (as stated at the end of the Abstract and in the Discussion). In the
context of CO2-induced climate change, an example of negative feedback would be an increase in
marine productivity, which would contribute to counterbalance the original disturbance (higher
pCO2). If, as a result of climate change, productivity decreases, then a positive feedback would be
occurring. But if the environmental changes considered (in this case, temperature and pCO2
combined) cause no effect on productivity, there is no feedback to speak of.
*Response:* We have modified negative feedback to 'no feedback' (see lines 593 & 603-604).

Lines 477-478 Many species have optimum temperatures higher than 20°C (Boyd et al. 2013
PlosOne, Chen 2015 J Plankton Res)
*Response:*  This has now been stated (see lines 436-438).

Lines 498-499 The logic behind this sentence is unclear: 'This may explain the lower PBm in the
combined treatment compared to elevated pCO2 and temperature individually'. If dinoflagellates have weaker CCMs compared to diatoms, how does that explain the fact that at the end of the
experiment PBm is smaller in the high CO2 treatment, where DIC is more abundant? How can you
connect a bulk property (PBm) with the abundance of dinoflagellates, which contribute a minor
fraction (<20%) of all phytoplankton? And how can you explain the effect of temperature? The
explanations of the antagonistic effect of temperature and pCO2 on phytoplankton biomass and
photosynthesis are confusing. This antagonistic effect is quite puzzling and, without trying to put
forward speculative mechanisms, the authors might just acknowledge that additional studies are
required to figure this out.

*Response:* This paragraph has now been edited to reflect your comments and make our assertions
clearer.
Lines 493-494 Such high pH values denote strong consumption of DIC which is associated with
nutrient depletion – hence slow growth rates likely result from nutrient limitation
*Response:* We have now acknowledged the role of nutrients in this sentence.

Line 560 Again, the sign of the potential feedback mechanism is wrong. If high CO2 leads to higher
photosynthesis and more CO2 uptake by the ocean, this represents a negative feedback, not a
positive one. It is a negative feedback because the outcome contributes to counterbalance or
neutralize the original disturbance (hence it is a stabilising mechanism).
*Response:* We have now corrected this statement to 'negative feedback'.

Line 581 There is no reduction, if productivity in the combined high CO2 and warm temperature
treatment was the same as in the control.

*Response:*  This has now been stated (see lines 603-604).
Figure 3, Table 1, also section 3.4. The POC:PON ratio cannot have units of mg m-3. This ratio should
be computed with molar concentrations, so that its units would be molC:molN.

*Response:* POC:PON has now been computed with molar concentrations and all associated sections,
table and figures have been updated.
Anonymous reviewer:
1) While limited resources or time are an understandable reason why some data may not be
available, you still have to discuss the scientific implications. For instance, what is the potential
error/uncertainty and would it affect your conclusions.

*Response:*  This has now been discussed
2) Phaeocystis was dominating the community towards the end of the experiment at high CO2,
which nanophytoplankton group was dominating at elevated temperatures? In this respect, it
appears a much more interesting/pressing question why phytoplankton biomass decreased in the
control and combined treatment towards the end (while the two others continued on their
trajectory) at nutrient replete conditions. Which direct or indirect effects could be responsible (grazing)?

*Response:* Nanophytoplankton were enumerated by flow cytometry and it was not possible to discern the species, except for Phaeocystis spp., from this. A statement has been added to qualify this (see lines 357-358) and differences in nanophytoplankton size classes between treatments were previously highlighted in the results section.

3) L345: A bracket seems to be missing.

*Response:* Now corrected.

4) L362: 'weilessii' should be in lower case

*Response:* Now corrected, although we have maintained the spelling as 'wailesii' as per the literature and text books.

5) L414-422: What is a 'variability in the C acquisition strategy', how would it be different to other species and where is the reference? Also, it appears not so much a community shift towards

Phaeocystis/nanophytoplankton at high CO2/high temperature but a switch towards a more diverse community in the control/combined treatments (also see my comment #1 and 2).

*Response:* We have now re-written this section of the paragraph

6) L430: It should read 'Chl a ratios'

*Response:* This has been corrected.

7) L430: Chlorophyll a is rather constant in comparison to the changes in biomass between day 25

and 36. Also, there is no data on an overall biomass decrease between T25 and T27.

*Response:* The data clearly shows a biomass decrease between T25 and T27

8) L433: Is there any evidence from the literature that Chl a to biomass/carbon ratios in these groups could explain your findings?

*Response:* We have now referenced this statement.

9) L439: There was no higher community biomass at elevated pCO2 in the Riebesell et al. (2007)

study. On this note the Delille et al. 2005 paper is on the same experiment. Furthermore, there was no significant CO2 effects on POC/PON in Riebesell et al. 2007 (L455). Finally, I am not aware of reports of Pbmax in the Riebesell et al. 2009 study.

*Response:* The reference to high biomass in Riebesell et al. (2007) has been removed (see line 468).

10) L478: There are definitely species which grow faster beyond temperatures of 20 degrees Celsius.

*Response:* This sentence has now been changed.

11) L564: The establishment of hypoxic zones would require eutrophic conditions.

*Response:* This has been clarified on lines 598-599.

12) L569: If there is no effect on Pbmax in the future treatment of combined temperature and CO2
increase in comparison to the control treatment which represents current conditions, then there are
also no feedbacks on atmospheric CO2.

*Response:* This has been corrected.

13) In the supplement, rather than showing absolute PAR measured in the incubation, you could
consider showing the ratio to that in air.

*Response:* This figure has now been revised.

14) Last but not least, you did not answer my question of how the flow cytometer was calibrated in
terms of size measurements.

*Response:* The calibration of the flow cytometry method has now been added to 'supplementary
material'.

[revised manuscript text omitted]

---

## Author Response (AR3)

**Author response to Associate Editor**

Associate Editor Decision: Publish subject to minor revisions (review by editor) (02

May 2018) by Emilio Marañón

Comments to the Author:

Dear authors,

Thank you for submitting a revised version of your manuscript. You have addressed adequately all the points that had been raised. However, there seems to be a problem with the reported values of the C:N ratio (Fig. 3F). Given the values of POC and PON

concentration, the molar C:N ratio should range roughly between 10-16. Please correct both the figure and the text describing it (lines 326-331). Also, the units of C:N could be simplified to the more commonly used molC:molN.

Best regards,

Emilio Marañón

*Response:* the POC:PON data has been corrected in **Fig. 3** and in the manuscript text from line 326 with the correct units (molC:molN)

[revised manuscript text omitted]